# Establishment of quantitative RNAi-based forward genetics in *Entamoeba histolytica* and identification of genes required for growth

Akhila Bettadapur[1], Samuel S. Hunter[2,¤a], Rene L. Suleiman[1], Maura C. Ruyechan[1], Wesley Huang[1], Charles G. Barbieri[3,¤b], Hannah W. Miller[1], Tammie S. Y. Tam[1], Matthew L. Settles[2,¤c], Katherine S. Ralston[1]*

**1** Department of Microbiology and Molecular Genetics, University of California, Davis, California, United States of America, **2** Genome Center, University of California, Davis, California, United States of America, **3** SeqMatic, LLC, Fremont, California, United States of America

☯ These authors contributed equally to this work.
¤a Current address: Invitae, San Francisco, California, United States of America
¤b Current address: Sestina Bio, Pleasanton, California, United States of America
¤c Current address: Medio Labs, Alameda, California, United States of America
* ksralston@ucdavis.edu

**Data Availability Statement:** Scripts and more details about the analysis are available at the GitHub repository associated with this publication:

## Abstract

While *Entamoeba histolytica* remains a globally important pathogen, it is dramatically understudied. The tractability of *E. histolytica* has historically been limited, which is largely due to challenging features of its genome. To enable forward genetics, we constructed and validated the first genome-wide *E. histolytica* RNAi knockdown mutant library. This library allows for Illumina deep sequencing analysis for quantitative identification of mutants that are enriched or depleted after selection. We developed a novel analysis pipeline to precisely define and quantify gene fragments. We used the library to perform the first RNAi screen in *E. histolytica* and identified slow growth (SG) mutants. Among genes targeted in SG mutants, many had annotated functions consistent with roles in cellular growth or metabolic pathways. Some targeted genes were annotated as hypothetical or lacked annotated domains, supporting the power of forward genetics in uncovering functional information that cannot be gleaned from databases. While the localization of neither of the proteins targeted in SG1 nor SG2 mutants could be predicted by sequence analysis, we showed experimentally that SG1 localized to the cytoplasm and cell surface, while SG2 localized to the cytoplasm. Overexpression of SG1 led to increased growth, while expression of a truncation mutant did not lead to increased growth, and thus aided in defining functional domains in this protein. Finally, in addition to establishing forward genetics, we uncovered new details of the unusual *E. histolytica* RNAi pathway. These studies dramatically improve the tractability of *E. histolytica* and open up the possibility of applying genetics to improve understanding of this important pathogen.

https://github.com/samhunter/Bettadapur-et-al-Entamoeba-histolytica-RNAi-library. All sequencing data from this project have been deposited in the Sequence Read Archive (SRA) database, under project number PRJNA672229: https://www.ncbi.nlm.nih.gov/bioproject/PRJNA672229.

**Funding:** DNA sequencing was carried out by the DNA Technologies and Expression Analysis Cores at the UC Davis Genome Center, supported by NIH Shared Instrumentation Grant 1S10OD010786-01. A.B. was supported by a fellowship from the UC Davis Training Program in Biomolecular Technology. M.C.R. was supported by the NIH T32 Comparative Medical Science Training Program, T32OD011147. T.S.Y.T. was supported by the Khaira Family Experiential Learning Award through the UC Davis College of Biological Sciences. This work was funded by NIH grants 1R01AI146914, 5R21AI154163 and a Pew Scholarship awarded to K.S.R. The funders had no role in study design, data collection and analysis, decision to publish, or preparation of the manuscript.

**Competing interests:** The authors declare that they have no competing interests.

## Author summary

*Entamoeba histolytica* is a globally important pathogen that is dramatically understudied. One of the major limitations of this organism is its challenging genome. RNAi is the state-of-the-art tool for genetic manipulation in *E. histolytica*, though the RNAi pathway has several noncanonical features. Here, we harnessed the RNAi pathway to enable RNAi-based forward genetics for the first time in this organism. We validated the RNAi library by performing the first *E. histolytica* RNAi screen and identified slow growth mutants. We showed that independently-generated mutants also exhibited slow growth phenotypes, and we characterized protein localization and domains for some of the identified slow growth genes. The RNAi library that we constructed enables modern, quantitative Illumina deep sequencing analysis to identify mutants that are enriched or depleted after selection. We developed a novel analysis pipeline to precisely define and quantify full-length gene fragments inferred from read mapping. Our approach differs from previous approaches for analysis of RNAi screens, and it better represents the actual DNA fragments and their quantities. This study dramatically improves the tractability of this important pathogen. Moreover, the strategies behind this RNAi library, and its analysis, are novel, and can be applied to other organisms.

## Introduction

*Entamoeba histolytica* is the causative agent of amoebiasis in humans. *E. histolytica* is common in developing nations and is responsible for ~50 million infections/year [1]. In an endemic area, approximately 80% of infants are infected with *E. histolytica* [2]. Malnutrition and stunting are associated with childhood *E. histolytica* infections [3]. In children enrolled in the Global Enteric Multicenter Study, *E. histolytica* infection was associated with the highest risk of death between enrollment and follow up [4]. Amoebiasis results in an estimated 15,500 deaths/year in children and 67,900 deaths/year in people of all ages [5]. There is no vaccine and therapy is limited [6]. The continued morbidity and mortality indicate that current treatment and prevention approaches are insufficient.

Despite its impact on human health, *E. histolytica* is dramatically understudied relative to other parasites. The genome is relatively difficult to manipulate. *E. histolytica* is tetraploid [7]. The genome contains many expanded gene families, which comprise 56% of the proteome [8,9]. The genome is ~75% A + T, which creates challenges for genomic approaches and has contributed to the relatively poor draft genome quality. The genome is ~20 Mb with ~8,000 genes [8,10]. For one third of genes, no homologues have been identified [10] and less than half of genes have gene ontology (GO) terms [8]. Thus, with limited sequence homology information and a difficult genome, functional approaches such as forward genetics are needed to identify genes that are relevant to pathogenesis.

Historically, the genetic tractability of *E. histolytica* has been limited. Forward genetics have not yet been established in *E. histolytica*, outside of two studies that used a plasmid-based over-expression library to screen for phenotypes [11,12]. More recently, the endogenous RNAi pathway in *E. histolytica* has been characterized, and exploited as a useful tool for targeted gene knockdown. Very recently, CRISPR/Cas9 has been established in *E. histolytica* [13], providing an important new tool for future genetic manipulation of this organism.

The existence of an RNAi pathway in *E. histolytica* that could be exploited for gene knockdown was implied by an approach that knocked down expression of amoebapore A [14]. However, the mechanism was unclear, and the expression of off-target genes was affected [15,16]. A

clonal line of stable transfectants carrying the amoebapore A plasmid was named the "G3" strain, and was subsequently used to knockdown additional genes in the context of an amoeba-pore A knockdown mutant background [15]. Other strategies for gene knockdown have included expression of genes in the antisense orientation [17], expression of long double stranded RNA [18], phagocytosis of bacteria expressing dsRNA [19], and expression of short hairpin RNA [20]. These techniques have been useful for reverse genetic studies, but not robust enough to be effective for every gene of interest.

In later studies, the *E. histolytica* RNAi pathway was characterized in detail. Small RNAs were found to be 27nt in length and 5' polyphosphorylated [21]. 5' polyphosphorylated siRNAs have only been seen in *E. histolytica* and *Caenorhabditis elegans* [22]. Supporting that these small RNAs are relevant to previous approaches that exploited gene silencing in *E. histolytica*, in the G3 strain, 27nt small RNAs mapped to amoebapore A, and when additional genes were silenced in this background, new small RNAs mapped to the newly silenced gene [23]. There is no obvious Dicer homologue, but Argonaute homologues and a potential noncanonical Dicer have been characterized [24,25].

Because endogenous small RNAs are polyphosphorylated, it was initially a challenge to exploit this pathway for gene knockdown. A major breakthrough was the discovery that RNAi knockdown could be induced by spreading of silencing [26]. A new "trigger" plasmid approach was established, which relies on endogenous small RNAs directed to a short frag-ment of an endogenously-silenced gene (called the trigger) to drive spreading of silencing to a fragment of gene of interest adjacent to the trigger [26]. This leads to long-term silencing of the gene of interest and the deposition of a repressive histone mark at the genomic locus [27]. This approach has been successfully applied in many studies, and in *E. invadens* [28–30].

In other parasites, RNAi knockdown has been successfully adapted for forward genetic approaches, and this has been very effective in uncovering genes relevant to parasite metabo-lism [31,32], DNA repair [33], virulence [34], transmission [35,36], and drug therapies [37–40]. Initially, these approaches relied on the generation of clonal lines after selection, and Sanger sequencing analysis of individual clones to identify the knocked down gene [31,32]. More recent studies used next generation sequencing on non-clonal mutant populations in order to identify mutants and quantify their relative abundance [33–40]. The latter approaches are useful because they can quantitatively identify mutants that are enriched by selection, and mutants that are less abundant or are lost after selection.

To enable forward genetics, we constructed and validated the first genome-wide *E. histoly-tica* RNAi knockdown mutant plasmid library. Its design allows for Illumina deep sequencing, and our novel analysis pipeline allows for quantitative identification of plasmids that are enriched or depleted after selection. Using a pilot version of the library, we performed the first RNAi screen in *E. histolytica* and identified slow growth mutants. These studies establish the first RNAi screen in *E. histolytica*, thereby dramatically improving the tractability of this pathogen.

## Results

### Approach for quantitative RNAi-based forward genetics

We devised an overall strategy for RNAi-based forward genetics in *E. histolytica* (Fig 1). For this approach, a library of gene fragments would be introduced into the *E. histolytica* "trigger" RNAi plasmid (pTrigger) [26], to create an RNAi library (Fig 1A). Amoebae would then be sta-bly-transfected with the library, and exposed to a selective pressure or screening strategy. Illu-mina deep sequencing analysis would then be used to sequence the inserts and to compare the selected amoebae to control, non-selected amoebae, in order to identify gene fragments that

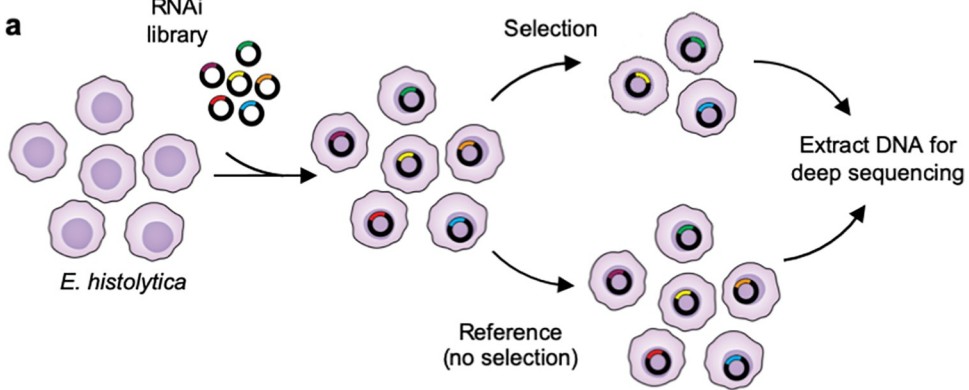

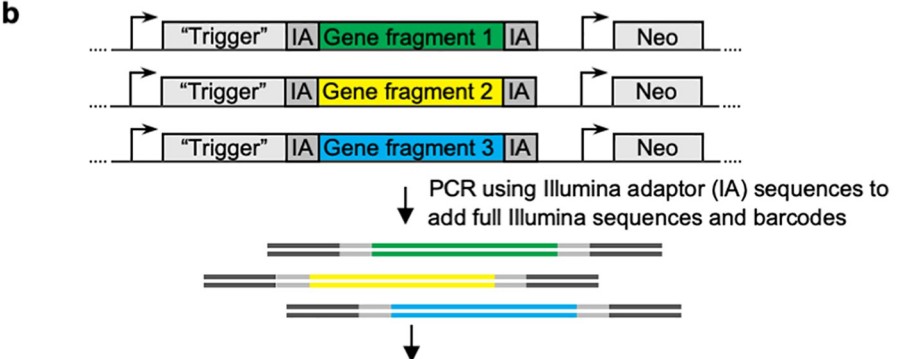

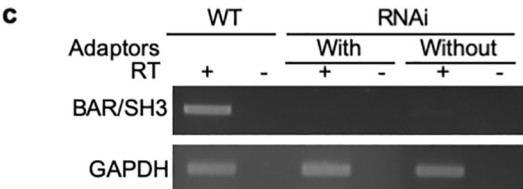

**Fig 1. RNAi library strategy. a,** The plasmid RNAi library contains random fragments of gDNA cloned into the trigger RNAi plasmid (pTriggerAdaptor). Different plasmid inserts are represented by different colors in this schematic. A heterogeneous population of stably transfected RNAi knockdown mutants can be exposed to selection or screening. To identify gene fragments that are represented in the population after selection/screening, gDNA is extracted for analysis. gDNA extracted from non-selected amoebae is used as a control. **b,** Schematic representation of the plasmid RNAi library. pTriggerAdaptor is shown, with different inserts represented by different colors. The "trigger" is a 132 bp fragment of EHI_048600. pTrigger was modified to create pTriggerAdaptor, containing custom Illumina TruSeq adaptor sequences (IA) upstream and downstream of the cloning site to facilitate amplicon sequencing. A single round of PCR is needed to add the remaining Illumina sequences (shown in dark gray) and barcodes necessary for Illumina deep sequencing analysis. **c,** An 812 bp fragment of a BAR/SH3 domain-containing protein (EHI_091530) was inserted into pTrigger with or without custom Illumina TruSeq adaptors flanking the cloning site. The BAR/SH3 domain-containing protein was in the forward orientation and in frame relative to the trigger sequence. Stably transfected heterogeneous mutants were assessed for knockdown of the targeted gene using RT-PCR, with (+) or without (-) reverse transcriptase. GAPDH (EHI_187020) was used as a loading control.

are quantitatively enriched or depleted. We empirically determined that the stable transfection efficiency with the backbone of pTrigger is ~1/1000 amoebae. This relatively high stable transfection efficiency suggested that transfecting enough amoebae to achieve genome-wide RNAi knockdown would be feasible.

The Illumina priming sequences that are necessary for sequencing are typically added to samples by using a nested strategy with two sequential PCR reactions. We modified the trigger plasmid in order to flank the insert with Illumina adaptor sequences, creating pTriggerAdaptor, so that after extracting amoebic genomic DNA (gDNA), only a single PCR step would be needed to generate DNA ready for sequencing (Fig 1B). Using fewer PCR reactions is expected to reduce the potential for PCR bias that can change the representation of sequences. An additional benefit of this modification is that the original sequences flanking the insert were A/T rich and were not optimal for primer design. The insertion of these Illumina sequences into the knockdown plasmid did not change knockdown efficacy (Fig 1C).

## Construction of a plasmid RNAi library

Building on these findings, we devised a strategy to construct a genome-wide plasmid RNAi library (Fig 2A). We reasoned that since ~50% of the genome is coding, constructing a gDNA library would be a practical approach. To generate the plasmid library, random fragments of gDNA were inserted into the pTriggerAdaptor plasmid. Library construction was initially done on a small scale, to create a "pilot" library, where plasmids were harvested from a limited number of bacterial colonies. Building on the results of the pilot, a second library was constructed on a larger scale, to make a "final" library that was expected to have genome-wide coverage.

Previous studies defined that an insert size $\geq$ 500 bp is sufficient for knockdown [26,41]. Larger insert sizes are likely to lead to more efficient knockdown, but fragments > 1 kb are problematic for Illumina sequencing. The pilot library was constructed using gDNA fragments between ~400–600 bp. Building on the robust Illumina sequencing results from the pilot library (described in detail below), we constructed the final library, where gDNA fragments were larger (between ~500–900 bp) in order to maximize the efficacy of RNAi knockdown.

To make the RNAi library, high quality, RNA-free, gDNA (S1A and S1B Fig) was sonicated and size-selected (S1C Fig). Y-shaped adaptors were ligated to the fragmented DNA to ensure that each fragment was flanked by sequences matching the trigger plasmid upstream and downstream of the cloning site, in order to enable subsequent Gibson cloning. Adaptor ligation was successful, with an appropriate size shift seen using bioanalyzer analysis (Fig 2B and 2C). Gibson cloning and plasmid preps were done *en masse* with pooled plasmids. To check cloning efficiency, individual colonies were obtained and analyzed. Restriction analysis demonstrated that the ligation efficiency was very high, since 63 out of 67 plasmids examined contained an insert (Fig 2D).

## The RNAi library has genome-wide coverage

Illumina sequencing was used to analyze the pilot and final libraries, and the fragmented gDNA that was used to construct each library. Notably, our approach for analysis of the sequencing data is novel and differs from previous approaches for analysis of RNAi screens [33–40] since all results presented here are based on full-length DNA-fragments inferred from read mapping results (S2 Fig), rather than on more traditional read mapping coverage or CDS overlap counting strategies. Our approach was designed to better represent the actual DNA fragments present in the plasmid library, instead of just the short ends of these fragments that were sequenced on the Illumina platform. This strategy allowed us to precisely define whether a fragment might target one or multiple genes, and enabled a clearer view of fragment based genome coverage. This approach will also allow us to readily quantify the representation of gene fragments in future RNAi screen experiments, in order to identify specific fragments that are particularly impactful in a particular screen.

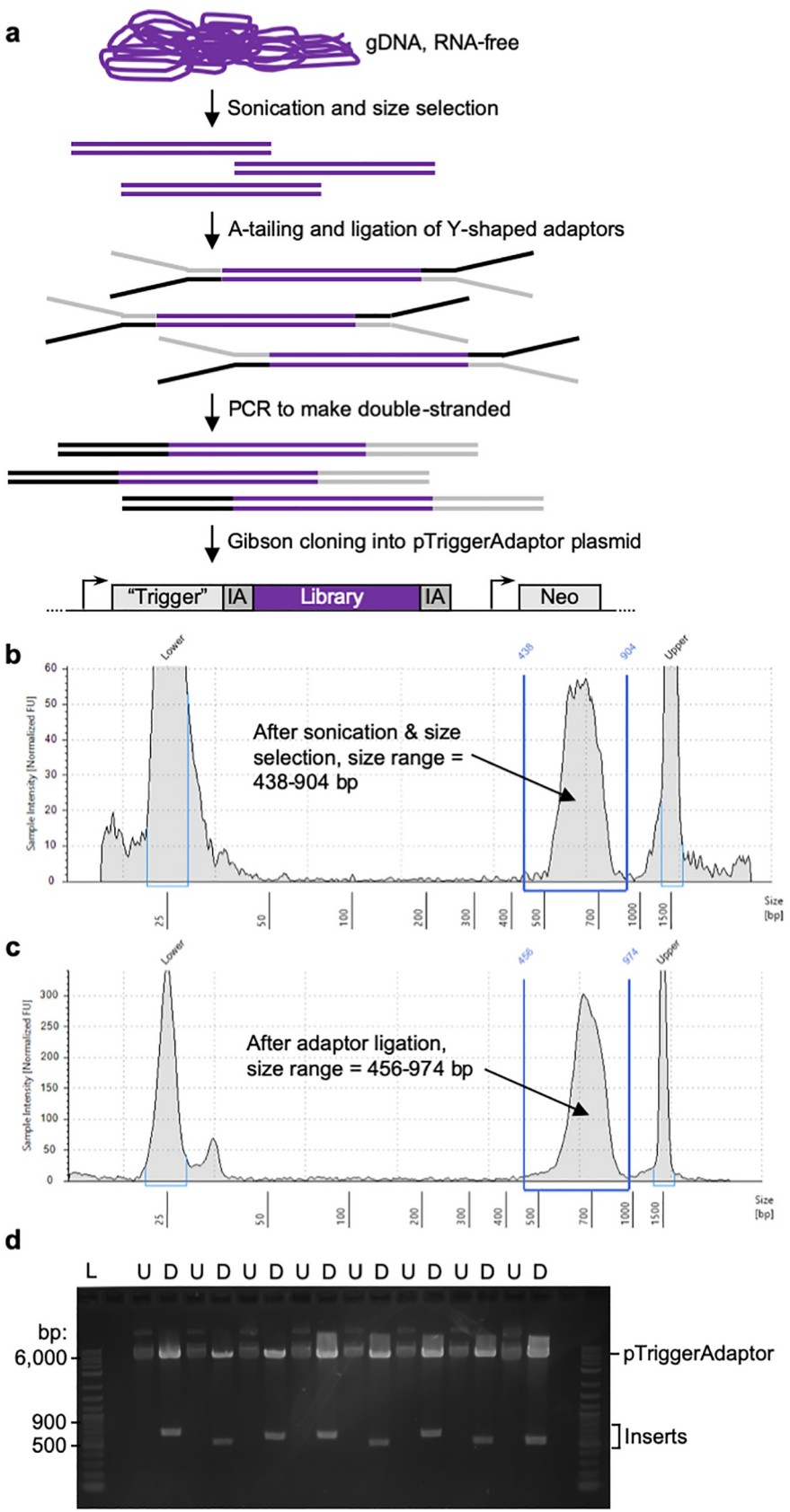

**Fig 2. RNAi library construction. a,** Steps to create the library. *E. histolytica* gDNA was extracted and treated with RNAse. Sonication, followed by gel purification, was used to isolate gDNA fragments. Fragments were end-repaired, A-tailed, and ligated to Y-shaped adaptors. The adaptors contain sequences corresponding to the region upstream and downstream of the cloning site in pTriggerAdaptor. The Y-shaped design ensures that, after PCR to make these sequences double-stranded, each gDNA fragment will be flanked by both the upstream/IA plasmid sequence (represented in black), and the downstream/IA sequence (represented in gray). Finally, the adaptor-ligated gDNA fragments were cloned into pTriggerAdaptor using Gibson cloning. **b–c,** Bioanalyzer traces show appropriate size shift after adaptor ligation, supporting that Y-shaped adaptors were successfully ligated. Example data are shown from optimization experiments that were performed to establish the protocols for library construction. **d,** Gibson cloning and plasmid preps were done *en masse* with pooled plasmids. To check cloning efficiency, individual colonies were obtained and analyzed. Shown are plasmid samples and restriction analysis from the pilot plasmid library preparation. Restriction analysis was performed with restriction enzymes that flank the cloning site (U, uncut; D, digested). 63 out of 67 plasmids that were analyzed contained an insert of the appropriate size range.

There did not appear to be bias in gDNA fragmentation or construction of the library, since the nucleotide and dinucleotide frequencies of the fragmented gDNA and the plasmids were the same as the genome (Figs 3A and S3A). Rarefaction curves and read fragment analysis showed that sequencing was performed to a sufficient depth to identify all unique fragments within the plasmid samples, but the same depth of sequencing was not deep enough to identify all unique fragments within the fragmented gDNA samples (S3B and S3C Fig).

Illumina sequencing analysis was performed on separate batches of the plasmid library that had been cloned on different days, and on technical sequencing replicates. Similar read counts per fragment were obtained from each batch and each sequencing replicate, showing that cloning and sequencing were reproducible and there was no evidence for bias within these steps (S4A and S4B Fig and S1 Table). Analysis of the sequencing results showed that the gDNA fragments that were used to create the pilot library had a mean size of 421 bp, and the fragments in the pilot plasmid library had a mean size of 427 bp (Fig 3B). The gDNA fragments that were used to create the final library had a mean size of 540 bp, and the fragments in the pilot plasmid library had a mean size of 555 bp (Fig 3B). The nearly identical size ranges of the gDNA fragments and cloned fragments further support that there was no bias during library construction.

Genome coverage plots of the final plasmid library over the 5 largest contigs in the genome visually showed that coverage of these contigs was complete, and the depth of coverage was consistent (Fig 3C). The breadth and depth of coverage over a zoomed in region of the largest contig is represented in Fig 3D. On a genome-wide level, the genome was well covered by fragments that were cloned into the library in both the forward and reverse orientation relative to the orientation of coding sequences (Fig 4 and Table 1). Supporting that the final library contained genome-wide coverage, 97.24% of the genome was represented (Table 1). 8307 of 8333 annotated genes were represented, and the majority (8301 genes) were represented by more than one unique fragment. Of the 26 genes that are missing, 22 are likely to be misannotated in the genome, since they were not seen in any sample, including fragmented gDNA (S5 Fig). Additionally, for the majority of these genes, there was no empirical evidence for gene expression in prior RNAseq or proteomic studies (S2 and S3 Tables). From these findings, we concluded that the final plasmid library has genome-wide coverage. We named the final library pEhKDL, since it represents an *E. histolytica* knockdown library.

## Forward genetic isolation of slow growing mutants

During the time that the final library was under construction, *E. histolytica* was stably transfected with the pilot library. To determine if the RNAi library could be used to identify genes that contribute to a phenotype, we screened for mutants with abnormal growth. To first ask how many different plasmids were present per stably transfected amoeba, clonal lines were

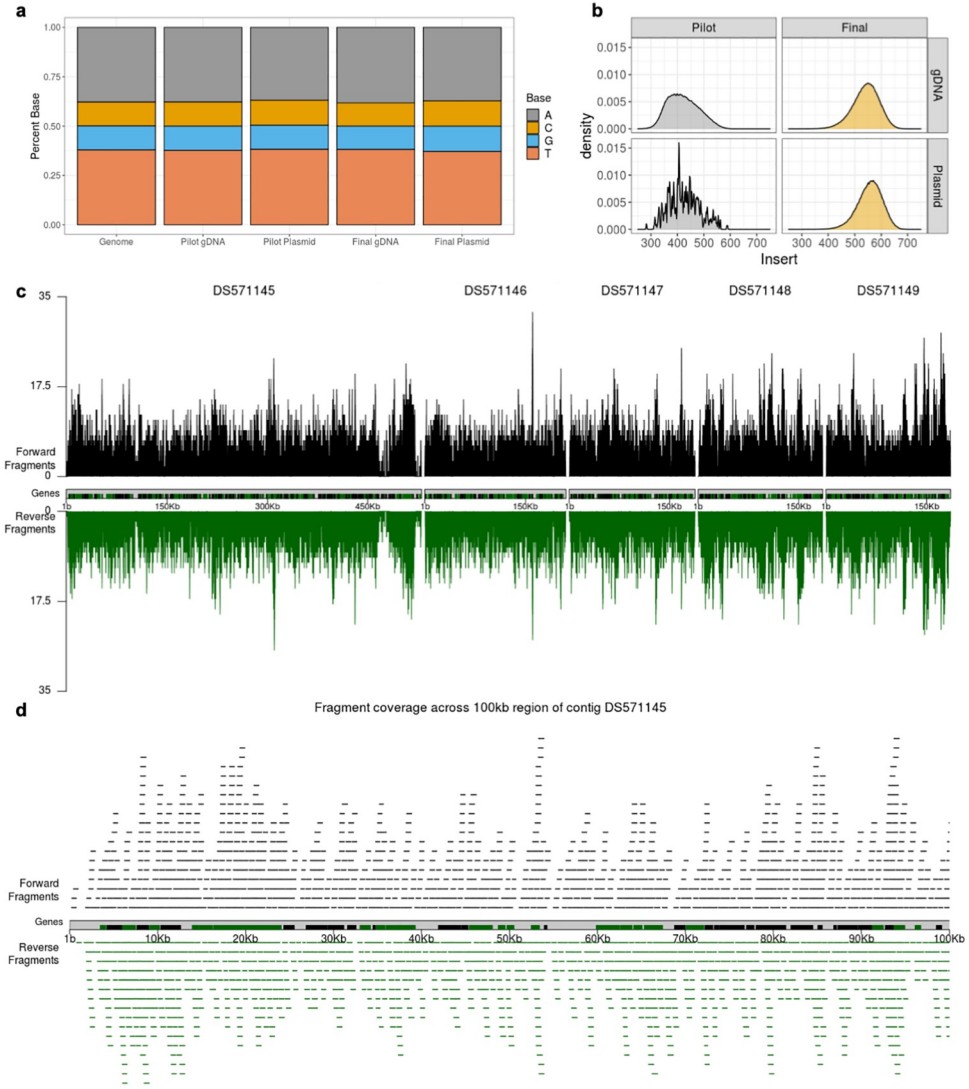

**Fig 3. The final RNAi library has genome-wide coverage. a,** Nucleotide frequency analysis of the HM1:IMSS reference genome, the pilot and final gDNA fragments, and the pilot and final plasmid libraries. Stacked bar plots show the percentage of each base. The nucleotide composition of the gDNA fragments and the final plasmid library is the same as the genome. **b,** Histograms showing the distribution of fragments sizes for the unique fragments in the pilot and final gDNA fragments, and the pilot and final plasmid libraries. Size in nucleotides is plotted on the X axis, and probability distribution density is plotted on the Y axis. **c,** Genome coverage plots of the final library, over the five largest contigs in the reference genome. The positions of unique fragments and the depth in fragments overlapping each genomic position are shown. The positions of genes are shown with black and green bars. Fragments that are in the same orientation as genes that are in the forward orientation in the reference genome are shown in black, and fragments that are in the same orientation as genes that are in the reverse orientation in the reference genome are shown in green. **d,** Visual representation of fragment coverage in the final library, over 100 kb of contig DS571145.

obtained using limiting dilution. Sanger sequencing analysis of clonal lines was consistent with only one unique plasmid (*i.e.*, one unique insert) per transfected amoeba (S6 Fig).

To screen for growth phenotypes, a selected number of clonal lines were analyzed. Slow Growth mutants 1 (SG1) and 2 (SG2) were isolated first and chosen for further characterization. These mutants had inserts corresponding to EHI_194310 and EHI_007010, which are annotated as a putative amino acid transporter and a hypothetical protein, respectively

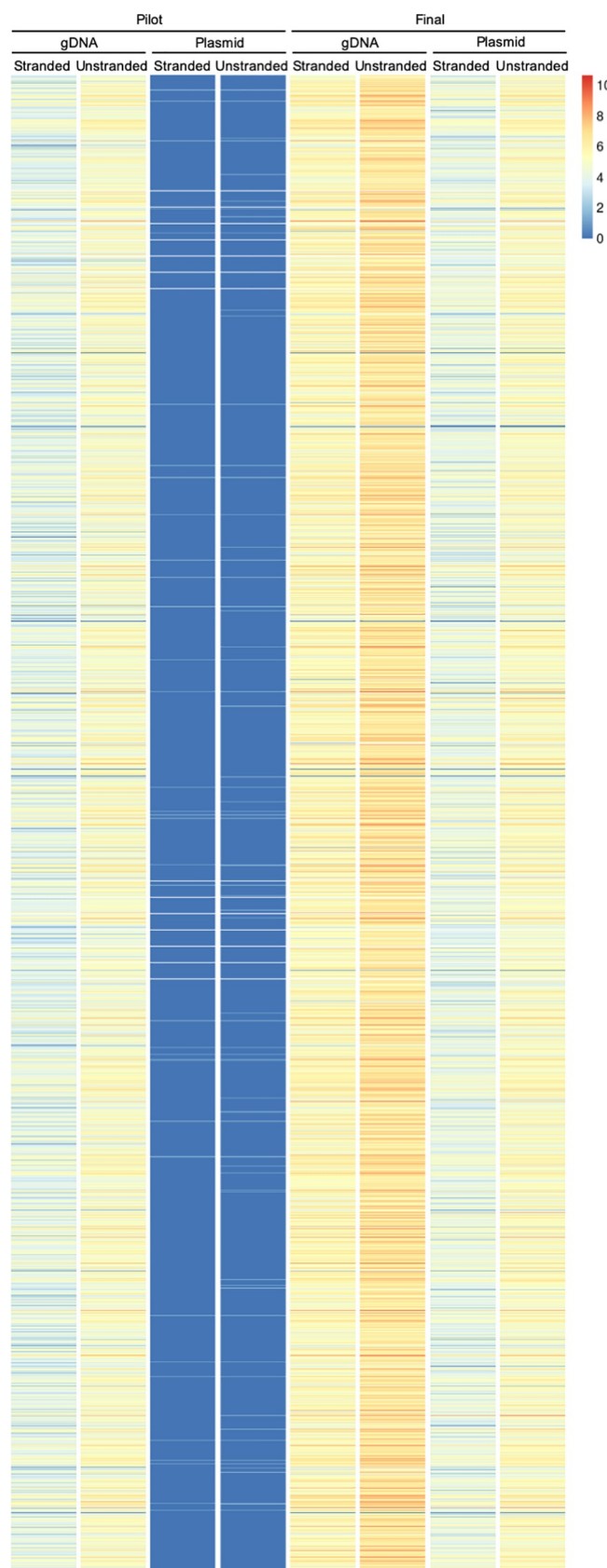

**Fig 4. Fragment directionality and coverage analysis.** Heat map showing the number of unique fragments per gene in the pilot and final gDNA fragments, and the pilot and final plasmid libraries. The number of unique fragments (Log$_2$) for each gene are plotted (color value, see legend). Stranded fragments are in the same orientation as genes. Unstranded fragments are the total number of fragments, both in the same orientation as genes and in the opposite orientation as genes.

(Table 2). These mutants had a statistically significant growth defect (Fig 5A). We have previously knocked down superoxide dismutase (EHI_159160) and determined that this results in a growth defect; thus, superoxide dismutase RNAi was used as a control. RT-PCR analysis showed that EHI_194310 and EHI_007010 were knocked down in the corresponding mutants (Fig 5B).

To determine if knockdown mutant phenotypes were reproducible, Gibson cloning was used to generate independent knockdown plasmids that contained a fragment of either EHI_194310 or EHI_007010, which differed from the fragment in the library mutants. *E. histolytica* was stably transfected with each of these new plasmids to create the independent knockdown mutants, SG1i and SG2i. Growth of heterogeneous, stable transfectants was measured, and these mutants had statistically significant growth defects (Fig 5C). RT-PCR analysis demonstrated that the corresponding target genes were knocked down (Fig 5D). The library mutants and independently generated mutants each exhibited statistically significant growth defects (Fig 5E). Thus, the phenotypes seen in library mutants were reproduced in independently generated mutants.

**Table 1. Library coverage analysis.** Analysis of coverage in the pilot and final gDNA fragments, and the pilot and final plasmid libraries. Genomic coverage (reported as bp covered, forward, reverse, and unstranded) was calculated using fragments of the appropriate orientation, inferred from mapped reads and the *coverage* function in the GenomicRanges R/Bioconductor package. The total number of covered bp are indicated, and the corresponding percentage of the HM1:IMSS reference genome that this represents. Forward fragments are in the same orientation as genes that are in the forward orientation in the reference genome. Reverse fragments are in the same orientation as genes that are in the reverse orientation in the reference genome. Stranded fragments are in the same orientation as genes, and are the combined total of both forward and reverse fragments. Unstranded fragments are the total number of fragments, both in the same orientation as genes in the reference genome and in the opposite orientation as genes. Genes that are targeted are those with at least one unique fragment with greater than or equal to 27nt that overlaps the gene. Genes that are missed are those that do not fit the criteria for targeted genes. Genes that are targeted by multiple fragments are those with two or more unique fragments. Counts were calculated using the countOverlaps function in the GenomicRanges package and annotation from AmoebaDB. The number of unique fragments is the total number of unique fragments in each sample.

| | Pilot | | Final | |
|---|---|---|---|---|
| | gDNA | Plasmid | gDNA | Plasmid |
| Covered bp, forward | 19961090 | 93257 | 20500538 | 19730276 |
| Covered %, forward | 96.0 | 0.5 | 98.6 | 94.9 |
| Covered bp, reverse | 19949512 | 99274 | 20499865 | 19734222 |
| Covered %, reverse | 96.0 | 0.5 | 98.6 | 94.9 |
| Covered bp, unstranded | 20401392 | 191674 | 20571421 | 20225883 |
| Covered %, unstranded | 98.1 | 0.9 | 98.9 | 97.2 |
| Genes targeted, stranded | 8299 | 152 | 8310 | 8302 |
| Genes missed, stranded | 34 | 8181 | 23 | 31 |
| Mean fragments per gene, stranded | 21 | 0 | 59.2 | 23.3 |
| Genes targeted by multiple fragments, stranded | 8270 | 54 | 8308 | 8276 |
| Genes targeted, unstranded | 8310 | 305 | 8310 | 8307 |
| Genes missed, unstranded | 23 | 8028 | 23 | 26 |
| Mean fragments per gene, unstranded | 42.9 | 0.1 | 119.9 | 47.2 |
| Genes targeted by multiple fragments, unstranded | 8307 | 111 | 8310 | 8301 |
| Number of unique fragments | 537066 | 621 | 1509696 | 489981 |

**Table 2. Identified slow growth mutants.** Summary of growth phenotypes of SG1 –SG12 slow growth mutants, and analysis of the sequences present in each corresponding library plasmid. Growth values represent cell numbers normalized to vector control transfectants on day 4 of growth assays. The corresponding raw data growth assay data are found in Figs 5A, S12A, S12C and S12E). Mutants exhibited statistically significant growth defects relative to vector control transfectants; *p*-values from ANOVA analyses are shown (see S12A, S12C and S12E Fig). The insert sizes are shown in bp. Insert directionality relative to the corresponding coding sequence in the annotated genome is shown. Insert frame relative to the trigger sequence in the pTriggerAdaptor is shown. The accession numbers for coding sequences that inserts correspond to are indicated, with available annotation information and GO terms for each gene (NA, not applicable). Predicted roles in metabolic pathways are shown; genes that are predicted to function in more than one metabolic pathway are indicated as "several." These predictions are based on data from KEGG Metabolic Pathways (https://www.kegg.jp), the MetaCyc Metabolic Pathway Database [43], and Chemical Entities of Biological Interest (ChEBI) database [44].

| Name | Normalized growth | *p*-value | Insert size (bp) | Insert direction | Insert frame | Accession | Annotation | Annotated domains | GO term | Metabolic pathways |
|---|---|---|---|---|---|---|---|---|---|---|
| SG1 | 16.74 +/- 2.17 | *** | 451 | Reverse | Out | EHI_194310 | Amino acid transporter, putative | Amino acid transporter, transmembrane domain | NA | NA |
| SG2 | 22.58 +/- 4.10 | ** | 544 | Forward | In | EHI_007010 | Hypothetical protein | NA | NA | NA |
| SG3 | 72.87 +/- 6.54 | * | 403 | Forward | Out | DS571580 | NA | NA | NA | NA |
| SG4 | 9.45 +/- 0.78 | **** | 351 | Reverse | Out | EHI_110390 | Transporter, major facilitator family | MFS transporter superfamily | NA | NA |
| SG5 | 14.50 +/- 0.91 | **** | 463 | Forward | In | EHI_024220 | Hypothetical protein | Importin beta family | Protein import into nucleus | NA |
|  |  |  |  | Forward | In | EHI_160920 | Hypothetical protein | Importin beta family | Protein import into nucleus | NA |
|  |  |  |  | Forward | In | EHI_171760 | Hypothetical protein | Importin beta family | Obsolete protein import into nucleus, docking | NA |
| SG6 | 15.19 +/- 2.23 | **** | 380 | Reverse | In | EHI_093960 | Pre-mRNA splicing factor, putative | Pre-mRNA-processing factor 6/Prp1/STA1 | mRNA splicing, *via* spliceosome | Several |
| SG7 | 19.59 +/- 1.89 | ** | 555 | Reverse | In | EHI_009580 | Tyrosine kinase, putative | Protein kinase-like domain superfamily | Protein phosphorylation | Several |
| SG8 | 65.35 +/- 5.05 | *** | 370 | Forward | In | EHI_148960 | Hypothetical protein | NA | NA | NA |
| SG9 | 11.08 +/- 0.96 | ns | 352 | Forward | In | DS571952 | NA | NA | NA | NA |
| SG10 | 11.45 +/- 1.29 | **** | 425 | Forward | In | EHI_069320 | C2 domain containing protein | C2 domain | NA | Several |
| SG11 | 40.47 +/- 6.77 | ** | 408 | Reverse | Out | EHI_098820 | Hypothetical protein | Dbl homology (DH) domain; Armadillo-type fold | Regulation of Rho protein signal transduction | NA |
| SG12 | 18.09 +/- 1.83 | ** | 322 | Reverse | In | DS571214 | NA | NA | NA | NA |

## Localization and domain analysis of proteins targeted in slow growth mutants

To further characterize SG1 and SG2, protein localization was examined. SG1 is annotated as a putative amino acid transporter (Table 2), with predicted transmembrane domains, but no predicted signal peptide sequence. SG2 is annotated as a hypothetical protein (Table 2), with no annotated domains. SG2 also lacks predicted transmembrane domains, predicted lipid modification sites, and a predicted signal peptide sequence. Thus, the localization of neither protein could be definitively inferred from sequence analysis.

We found that SG1 was localized to the amoeba surface. SG1 was also detected intracellularly, with a punctate distribution (Figs 6A, 6B, S7 and S8). Since these transfectants were heterogeneous, there was heterogeneity in the intensity of SG1 staining. The expression level of

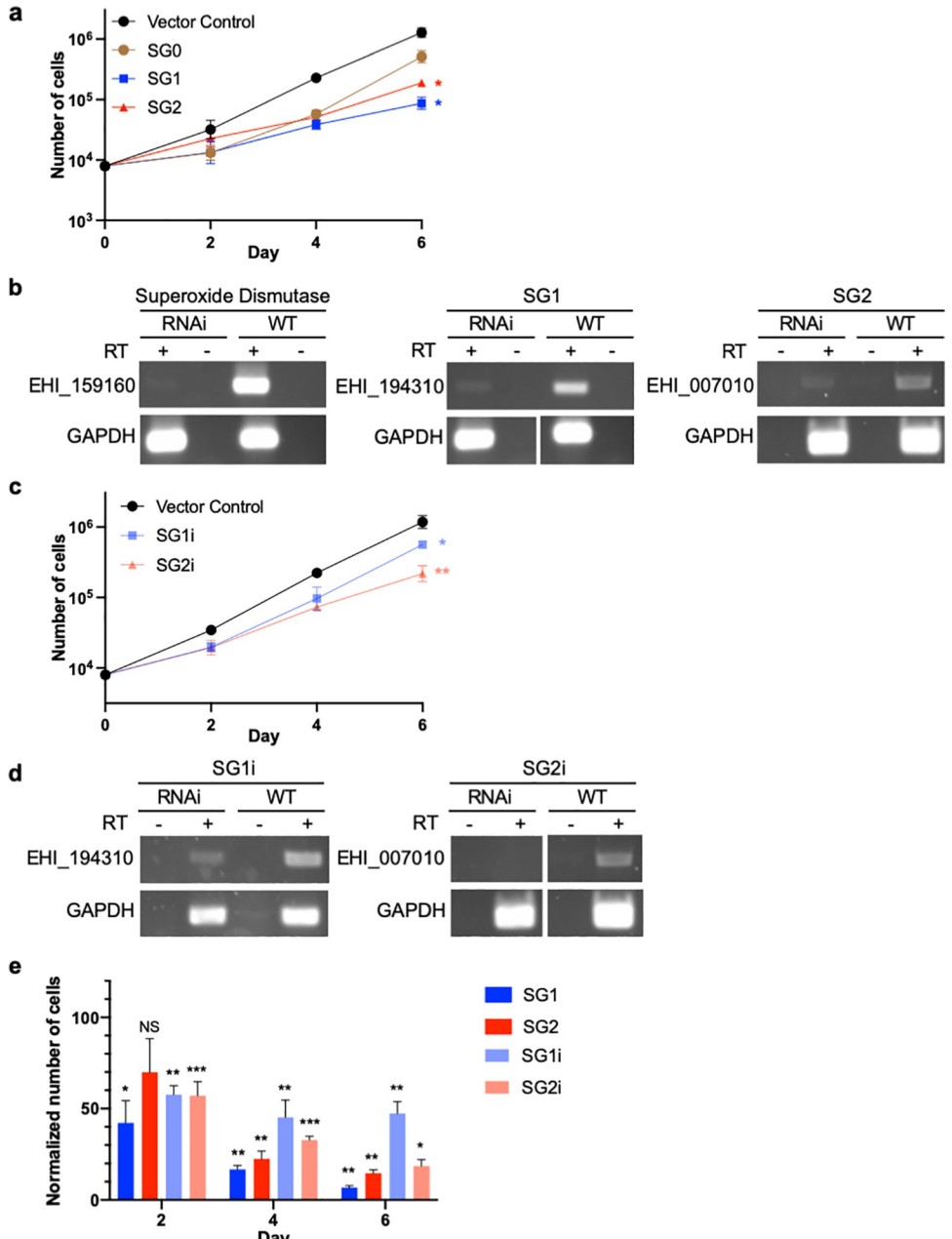

**Fig 5. Identification of slow growth mutants, with phenotypes that are reproducible in independently-generated knockdown mutants. a,** Growth analysis of clonal knockdown mutant lines obtained from screening the heterogeneous library transfectants for growth defects. Amoebae transfected with pTriggerAdaptor (vector control) were used as a negative control, and superoxide dismutase ("SG0"; EHI_159160) knockdown mutants were used as a positive control for slow growth. The library mutants SG1 and SG2 each exhibited a significant growth defect when compared to vector control transfectants. **b,** RT-PCR analysis demonstrates that the genes corresponding to the plasmid inserts in SG1 and SG2 (EHI_194310 and EHI_007010, respectively), were knocked down. **c,** Knockdown mutants were independently generated by using fragments of these genes that differed from the fragments in the library plasmids. These independent mutants are referred to as SG1i and SG2i. Growth analysis was performed using heterogeneous stable transfectants, and growth mutants exhibited a significant growth defect when compared to vector control. **d,** RT-PCR analysis demonstrates that EHI_194310 and EHI_007010 were knocked down in SG1i and SG2i mutants, respectively. **e,** Growth phenotypes in SG1, SG2, SG1i and SG2i. Growth values for each mutant were normalized to vector control transfectants. Mutants exhibited statistically significant growth defects relative to vector control transfectants.

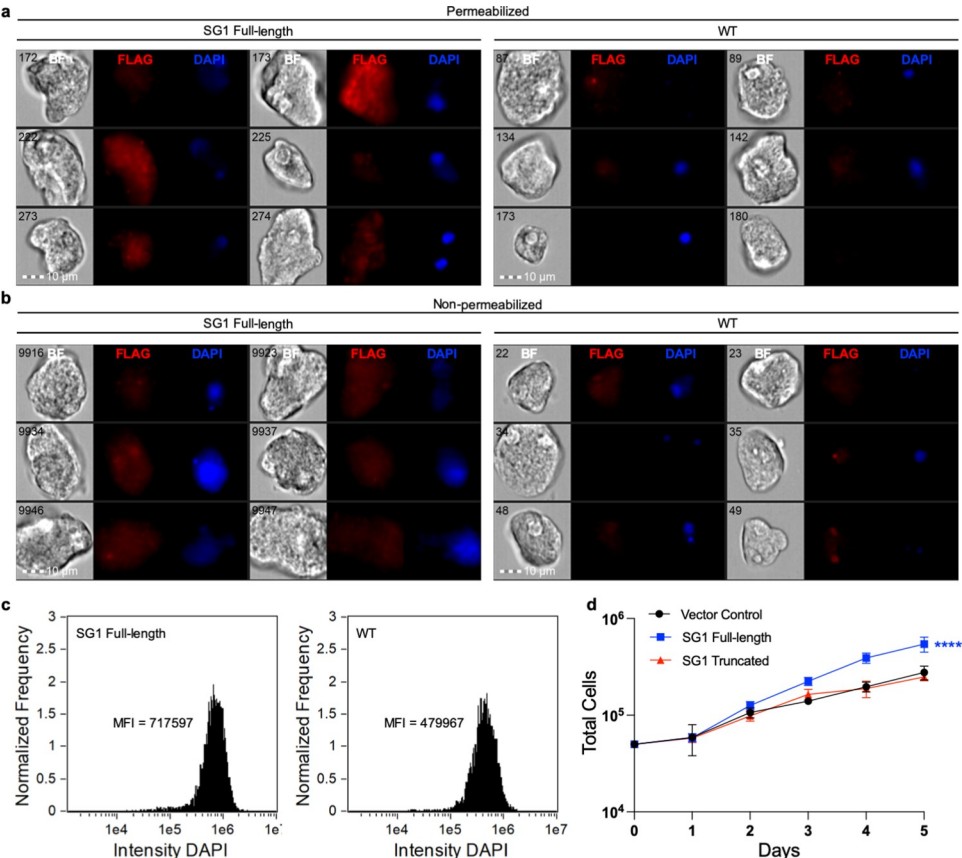

**Fig 6. SG1 localizes to the cytoplasm and cell surface, and overexpression of SG1 enhances growth.** Amoebae were stably transfected with a plasmid for expression of full-length FLAG-tagged SG1. Immunofluorescence was used to determine the localization of SG1 in heterogeneous transfectants and imaging flow cytometry was used for analysis. **a,** Localization of SG1 in FLAG-SG1 transfectants (left panel) and control staining of wild-type cells (right panel). Samples were permeabilized prior to antibody staining. Six random cells are shown for each condition. Shown from left to right are bright field images (BF), rabbit anti-FLAG antibody staining (FLAG, red), and DAPI staining (DAPI, blue). The numbers in the BF images indicate the object/image number. **b,** Localization of SG1 in FLAG-SG1 transfectants (left panel) and control staining of wild-type cells (right panel), as in panel A, except samples were not permeabilized. **c,** Histograms showing the intensity of the DAPI staining in images collected from FLAG-SG1 transfectants (left panel) or wild-type cells (right panel). The mean fluorescence intensity (MFI) is indicated on each histogram. **d,** Growth analysis of heterogeneous transfectants expressing full-length SG1 or a C-terminal SG1 truncation mutant. Amoebae transfected with pEhEx (vector control) were used as control.

SG1 was very low (S8 Fig), and was dramatically lower than the expression level of a control, endogenous surface protein (S9 Fig). Interestingly, amoebae overexpressing full-length SG1 consistently exhibited brighter DAPI staining (Fig 6C) and/or more dividing cells with two nuclei, compared to wild-type amoebae (Fig 6A and 6B), suggesting that proliferation was higher in these amoebae. Growth analysis showed that amoebae overexpressing full-length SG1 grew faster than vector control transfectants (Fig 6D). To gain a deeper understanding of domains required for SG1 function, a C-terminal truncation mutant was overexpressed, and since it did not lead to increased growth, we concluded that this truncation interfered with protein function (Fig 6D).

Unlike SG1, SG2 was not detectable on the amoeba surface (Figs 7A, 7B, S7 and S10). Instead, SG2 exhibited staining throughout the amoeba cytoplasm (Figs 7A and S10). Since these transfectants were heterogeneous, there was heterogeneity in the intensity of SG2

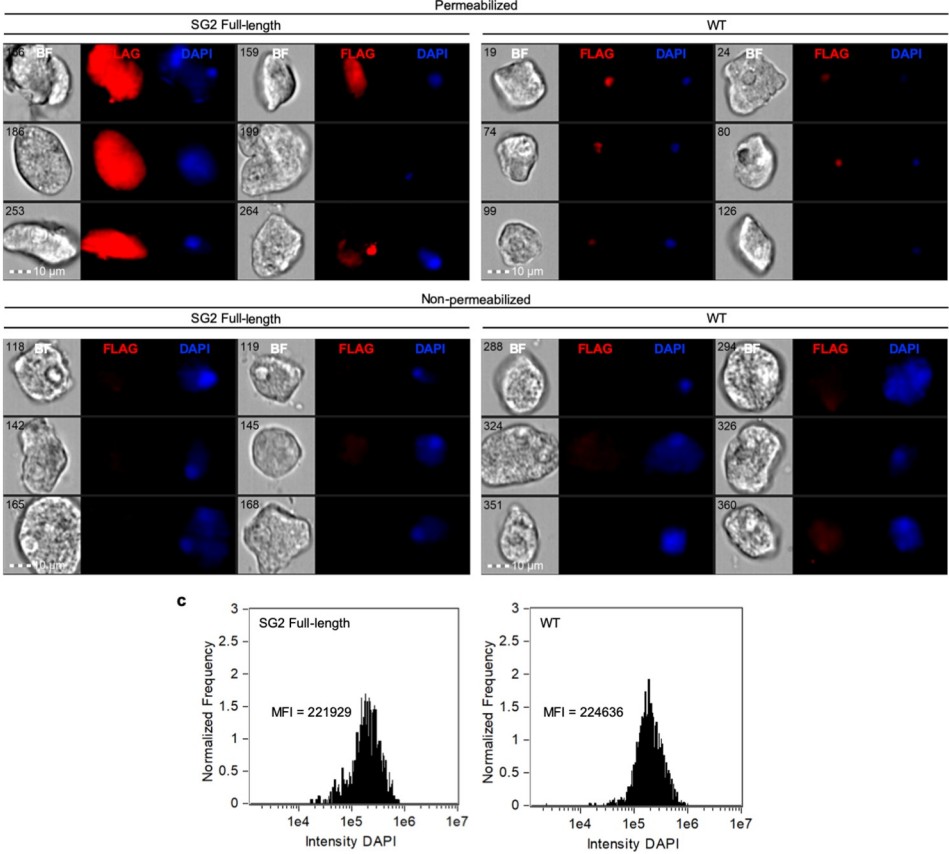

**Fig 7. SG2 localizes to the cytoplasm.** Amoebae were stably transfected with a plasmid for expression of full-length FLAG-tagged SG2. Immunofluorescence was used to determine the localization of SG2 in heterogeneous transfectants and imaging flow cytometry was used for analysis. **a,** Localization of SG2 in FLAG-SG2 transfectants (left panel) and control staining of wild-type cells (right panel). Samples were permeabilized prior to antibody staining. Six random cells are shown for each condition. Shown from left to right are bright field images (BF), mouse anti-FLAG antibody staining (FLAG, red), and DAPI staining (DAPI, blue). The numbers in the BF images indicate the object/image number. **b,** Localization of SG2 in FLAG-SG2 transfectants (left panel) and control staining of wild-type cells (right panel), as in panel A, except samples were not permeabilized. **c,** Histograms showing the intensity of the DAPI staining in images collected from FLAG-SG2 transfectants (left panel) or wild-type cells (right panel). The mean fluorescence intensity (MFI) is indicated on each histogram.

staining. Like SG1, the expression level of SG2 was very low (S10 Fig), and was dramatically lower than the expression level of a control, endogenous surface protein (S9 Fig). Overexpression of full-length SG2 was not associated with increased growth, and no difference was detected in the intensity of DAPI staining between SG2 overexpressors and wild-type cells (Fig 7C). Unlike the rabbit antibody used to localize SG1 (Fig 6A and 6B), the mouse antibody used to localize SG2 was associated with some background nuclear staining in wild-type cells (Figs 7A, 7B, S7 and S11). By contrast, nuclear staining was not pronounced in SG2 overexpressors (S11 Fig).

## Bioinformatic analysis of slow growing mutants

From the pilot RNAi library transfection, ~64 additional clonal lines were obtained and 10 additional clonal lines with statistically significant growth defects were identified (S12 Fig and Table 2). The insert present in each clonal line was identified by Sanger sequencing, and the results were again consistent with one unique plasmid per transfected amoeba. Inserts ranged

in size from 322 to 555 bp. In previous studies that used pTrigger for gene silencing, the gene of interest for knockdown was in the forward orientation and in frame with the trigger [26,28–30,41,42]. In the slow growth mutants, inserts were in the forward or reverse orientation, and in or out of frame (Table 2). Inserts that were in the reverse orientation and out of frame, like SG1, nonetheless led to measurable knockdown (Fig 5B). Both the directionality of the gene of interest, and the frame, are therefore dispensable for knockdown.

Nine of the twelve slow growth mutants contained an insert that mapped to a gene (Table 2). The three mutants with inserts that did not clearly map to an annotated gene (SG3, SG9 and SG12) did map to potentially unannotated ORFs, and also partially mapped (*i.e.*, contained stretches of homology) to annotated genes. BLAST was used to search for homologs of the ORFs that mapped to SG3, SG9, and SG12, and homologs were found in *E. dispar* for each ORF (Expect values of 4e -31, 1e -10, 1e -9, respectively), suggesting that these ORFs are genes that were missed in the annotation of the *E. histolytica* genome. The *E. histolytica* genome contains a very large number of gene families [8,9], evidencing a high level of functional redundancy. One mutant, SG5, contained an insert that mapped to more than one gene (Table 2). Due to the level of identity between the multiple hits, multiple related genes may be knocked down in this mutant.

Several mutants contained inserts that mapped to genes with annotated functions consistent with roles necessary for cellular growth, such as an amino acid transporter, a major facilitator transporter, and a splicing factor (Table 2). Three mutants contained inserts with predicted roles in metabolic pathways, based on data from KEGG Metabolic Pathways (https://www.kegg.jp), the MetaCyc Metabolic Pathway Database [43], and Chemical Entities of Biological Interest (ChEBI) database [44] (Table 2). Several mutants contained inserts that mapped to genes annotated as hypothetical, or that lacked annotated domains (Table 2). Thus, in addition to validating the RNAi library approach, the 12 slow growth mutants that were uncovered shed light on genes that are required for *E. histolytica* growth and emphasize how forward genetics can be used to assign functional information to hypothetical proteins.

## Discussion

We constructed and validated the first genome-wide *E. histolytica* RNAi knockdown library. We performed a screen for slow growth mutants and identified 12 new mutants, providing a proof-of-concept that demonstrates the library can be used to identify mutant phenotypes. Independent knockdown mutants phenocopied the growth defects seen in the library mutants, showing that library mutant phenotypes are reproducible. Thus, our studies both identify genes that are critical for growth, and establish RNAi-based forward genetics in *E. histolytica* for the first time. The library represents a valuable new tool for *E. histolytica*, that can now be applied towards the study of many processes.

Slow growth mutants were identified among clonal knockdown mutant lines. 12 mutants were identified among ~80 total clonal lines. Among the identified genes targeted in these mutants, many genes had annotated functions consistent with roles in cellular growth, and/or had predicted roles in metabolic pathways, based on available metabolic pathway databases (https://www.kegg.jp/) [43,44]. No genes were identified for which a role in cellular growth would be implausible. Several of the identified genes were annotated as hypothetical, or lacked annotated domains. All of the gene hits in our screen had previous evidence of gene expression in RNAseq analyses [45,46]. Thus, the slow growth phenotype begins to provide functional information to genes for which existing annotation does not inform understanding of protein function. All of the annotated gene hits in our screen had homologues in the related human-infectious *Entamoeba* species *E. dispar*, *E. nuttali* and *E. moshkovskii*. Thus, this type of screen

may be informative in identifying potential drug targets that are conserved among human-infectious *Entamoeba* species.

In our analysis of slow growth mutants, we seeded cells at a set starting concentration, and then measured the number of cells at various time points thereafter. We defined slow growth phenotypes as those that resulted in significantly lower cell numbers over time, relative to control cells. In the case of cells overexpressing full-length SG1, an increased growth phenotype was also associated with a higher intensity of DAPI staining compared to control amoebae. The assays that we used do not formally distinguish between an altered rate of proliferation and altered longevity of individual cells in culture.

The *E. histolytica* genome contains numerous gene families [8,9]. An advantage of RNAi *vs.* alternative approaches (*e.g.*, gene knockouts) is the potential to simultaneously interfere with multiple gene family members. RNAi is similarly advantageous for genomes that are polyploid, which is also relevant to the tetraploid [7] *E. histolytica* genome. We identified one mutant that contained an insert that mapped to more than one gene. With the high levels of sequence identity, it is likely that multiple genes are knocked down in this mutant. We were able to identify a slow growth phenotype in this mutant, which suggests that the RNAi library may be effective in identifying mutant phenotypes even in the context of high levels of functional redundancy.

We have used pTrigger to knockdown many genes in our laboratory, and, in general, have been able to recover knockdown mutants for essentially every targeted gene. Compared to clonal lines, heterogeneous transfectants have typically had more variable levels of knockdown, and sometimes appear to have lower levels of knockdown [47]. We obtain clonal lines by limiting dilution, but another effective available method is cloning by isolating *E. histolytica* colonies on soft agar plates [48]. Among clonal lines obtained from heterogeneous transfectants, we have seen wide variation, including clones with high levels of knockdown and clones that lack knockdown. The SG1 and SG2 clonal lines initially obtained from the library appeared to have an equal or higher level of knockdown than the heterogeneous independent SG1i and SG2i mutants that targeted the same gene. The heterogeneous independent knockdown mutant SG1i had a less severe growth phenotype than the clonal library mutant SG1, which may be due to variation in knockdown in the heterogeneous population. Loss of gene silencing (and loss of a growth phenotype) over time was noted in the heterogeneous SG2i mutants, consistent with the selective disadvantage of the growth phenotype. Related to this point, when a selection is applied to a library mutant population, it will be important to compare selected and non-selected controls, in order to differentiate mutants that are lost due to a growth disadvantage *vs.* mutants that are lost due to selection.

The library was designed to facilitate Illumina deep sequencing analysis in order to quantitatively identify gene fragments that are enriched or lost after selection. We constructed pilot and final versions of the library. Multiple lines of evidence supported that there was no bias in library construction, including nucleotide and dinucleotide frequencies, and size comparisons between the gDNA fragments and fragments in the final plasmids. Libraries were prepared in batches that were cloned on different days. Similar read counts per fragment were obtained from batches, again supporting that there was no bias in cloning or sequencing analysis. The majority of the genome (97.24%) was covered in the final plasmid library, and the majority of genes were represented (8307 out of 8333 annotated genes). 26 annotated genes are not represented. Of these, 22 genes were not seen in any sample, including gDNA, suggesting that they represent errors in annotation. Consistent with this, there was no published evidence for gene expression for the majority of these genes. The 4 remaining genes that were not seen in the final library had some of the lowest sequencing coverage in the fragmented gDNA sample, suggesting that they were missed due to lack of abundance in the fragmented gDNA that was used

for library construction. Thus, with 8307 out of 8311 correctly annotated genes represented, and 97.24% of the genome, we concluded that the library represented full genome coverage.

A future variation on this approach would be to make a library using fragmented cDNA instead of gDNA. We reasoned that since ~50% of the genome is coding, gDNA is a reasonable approach, since ~50% of the plasmids in the library would lead to gene silencing. A limitation of a gDNA library is that some sequences such as rRNA and tRNA, which are abundant in the genome, are represented in the library, and are not useful for identification of mutant phenotypes. A cDNA library would remove these sequences, but would carry other limitations. The most highly expressed genes would be overrepresented in a cDNA library, and genes that are not expressed in the conditions in which cDNA was obtained would not be represented in the library. The lower abundance of material for fragmentation and cloning is also a consideration. mRNA is a very small proportion of total RNA, and in our initial attempts to prepare cDNA for library construction, insufficient material was recovered. With more initial total RNA, a cDNA library could be constructed in the future, and would be complementary to the gDNA library that was constructed here.

We asked whether the frame or the directionality of the insert was critical for knockdown using the trigger plasmid. In previous studies, the gene of interest for knockdown was in the forward orientation and in frame with the trigger [26,28–30,41,42]; thus, the frame and orientation requirements had not been tested experimentally. Among the slow growth mutants that we identified, inserts were in the forward or reverse orientation, and in or out of frame. Note that the adaptor sequence that was inserted between the trigger and the insert does not change the frame. Inserts that were in the reverse orientation and out of frame still led to knockdown. Both the directionality of the gene of interest, and the frame, are therefore dispensable for knockdown. Since the RNAi pathway in *E. histolytica* is unusual and mechanistic details are still coming into focus, our results add to understanding of this pathway. The lack of orientation and frame requirements for knockdown is useful, since gDNA fragments are random and thus are ligated into the library plasmid in either orientation and in any frame. Previous studies had defined that a gene fragment $\geq$ 500 bp is sufficient for knockdown [26,41]. We found that fragments as small as 322 bp were sufficient to result in slow growth phenotypes, suggesting that efficient knockdown is possible with smaller fragments.

*E. histolytica* is dramatically understudied relative to other parasites, even though it heavily impacts human health, causing significant morbidity and mortality. The RNAi library that we developed dramatically improves the tractability of *E. histolytica*. In other parasites, RNAi screens have uncovered genes relevant to molecular and cellular biology, virulence and therapeutics. We look forward to the new information about *E. histolytica* that can be gleaned by applying RNAi screens in the future. Since the design of this RNAi library allows for quantitative identification of gene fragments that are enriched or depleted after selection, we expect that this library can be used to identify genes involved in a wide variety of processes, including essential genes that are relevant to the development of new therapies.

## Materials and methods

### Cell culture

*E. histolytica* HM1:IMSS (ATCC 30459) trophozoites were cultured in TYI-S-33 media, supplemented with 15% heat inactivated Adult Bovine Serum (Gemini Bio Products), 80 units/ml penicillin and streptomycin (Gibco), and 2.3% Diamond Vitamin Tween 80 solution 40x (Sigma Aldrich), at 35˚C. Amoebae were harvested when flasks were approximately 80% confluent.

## gDNA extraction

*E. histolytica* gDNA was harvested using the Quick-DNA miniprep kit (Zymo), cetyl trimethylammonium bromide (CTAB) extraction, or the Wizard Genomic DNA Purification kit (Promega). The Quick-DNA miniprep kit (Zymo) was used according to manufacturer's instructions for cell suspension samples. For CTAB extraction, cells were washed and then lysed at 55°C for 1 hour in a solution of 0.1 M EDTA with 0.25% SDS and 0.1 mg/ml proteinase K. NaCl was added to a final concentration of 0.7 M, pre-warmed CTAB was added to a final concentration of 1%, and samples were incubated for 20 minutes at 65°C. An equal volume of chloroform was added, and samples were centrifuged at 20,600 x *g* for 10 minutes. Supernatants were transferred into fresh tubes, an equal volume of phenol:chloroform:isoamyl alcohol (25:24:1) was added, and samples were centrifuged at 20,600 x *g* for 10 minutes. Supernatants were transferred into fresh tubes, DNA was precipitated with 2.5 volumes of 100% ethanol at room temperature for 5 minutes, and samples were centrifuged at 20,600 x *g* for 10 minutes. DNA pellets were washed in 70% ethanol. DNA was left to dry at room temperature and dissolved in water. For DNA harvested using the Quick-DNA miniprep kit or CTAB extraction, RNAseIF (NEB) was used to remove RNA contamination, according to manufacturer's instructions. The Wizard Genomic DNA Purification kit (Promega) was used according to the manufacturer's instructions for tissue culture cells. Because DNA harvested using the Wizard Genomic DNA Purification kit was of higher quality, gDNA for plasmid library construction was prepared using this kit.

## Plasmids and cloning

The trigger plasmid [26] contains 132 base pairs of the trigger gene (EHI_048600) in the pKT3M expression plasmid backbone [49]. Q5 Site Directed Mutagenesis (NEB) was used to insert a SmaI restriction enzyme site after the trigger for cloning purposes, to create "pTrigger". To create knockdown plasmids to target individual genes, pTrigger was linearized using SmaI (NEB) and dephosphorylated using rSAP (NEB). A fragment of the gene of interest was amplified from gDNA using PCR primers that contained homology to pTrigger, and the PCR product was then cloned into the backbone using the Gibson Assembly Ultra kit (Synthetic Genomics Inc.), following the manufacturer's instructions. See S4 Table for the sequences of primers that were used during plasmid construction.

To test if the Illumina adaptor sequences interfere with knockdown, pTrigger plasmids were created with and without adaptors. First, a pTrigger plasmid targeting a BAR/SH3 domain-containing protein was created by inserting 812 bp of EHI_091530 (147–958). This plasmid was further modified using Q5 Site Directed Mutagenesis (NEB) to add the custom adaptors ACACTCTTTCCCTACACGACGCTCTTCCGATCT and GATCGGAAGAGCAC ACGTCTGAACTCCAGTCAC, which are compatible with Illumina TruSeq sequencing primers, upstream and downstream of the insert, respectively. After determining that the presence of the Illumina adaptors did not interfere with knockdown, this construct was modified to remove the BAR/SH3 domain-containing protein and replace it with a SmaI restriction site for further cloning purposes, to create pTriggerAdaptor. pTriggerAdaptor was used as the backbone for plasmid library generation.

Superoxide dismutase knockdown mutants were generated by inserting 597 bp of EHI_159160 (1–597) into pTrigger. Independent knockdown mutants (SG1i) of SG1 were generated by inserting 525 bp of EHI_194310 (32–556) into pTrigger. Independent knockdown mutants (SG2i) of SG2 were generated by inserting 487 bp of EHI_007010 (59–545) into pTrigger.

To create the N terminal FLAG-tag constructs, Q5 Site Directed Mutagenesis (NEB) was used to insert a 3X FLAG epitope tag sequence preceding an XhoI restriction site in the pEhEx expression plasmid backbone [49], to create pEhEx-FLAG. For localization of SG1 (EHI_194310) and SG2 (EHI_007010), each individual full-length gene was cloned into pEhEx-FLAG, to create pEhEx-FLAG-SG1 and pEhEx-FLAG-SG2, respectively. For expression of truncated SG1, nt 1010–1527 were omitted, and the resulting truncated gene was cloned into pEhEx-FLAG, with a stop codon added at the 3' end of the truncated sequence.

## Plasmid library construction

*E. histolytica* gDNA was harvested using the Wizard Genomic DNA Purification kit (Promega), following the manufacturer's instructions for tissue culture cells. RNAse treatment is included in this protocol. Quality and size of gDNA was verified by Qubit and gel electrophoresis. gDNA was further purified and concentrated using TailorMag Purification beads (SeqMatic), and then sheared using a Covaris M220 sonicator. For size selection, sheared gDNA was loaded into three lanes on a 6% TBE gel. The gel was stained using SYBR Safe DNA Gel Stain (Invitrogen), and three bands, covering from ~400 –~600 bp (for the pilot library) and ~500 –~900 bp (for the final library) were extracted from the gel using the TailorCut Gel Cutter and Gel Breaker system (SeqMatic). gDNA was purified using TailorMag Purification beads. The sizing and concentration of the purified material was determined by TapeStation (Agilent). Purified gDNA corresponding to the second and third gel slices from each lane was combined. gDNA was further purified and concentrated using TailorMag Purification beads, and then processed using a TruSeq XT library prep kit (Illumina). All steps followed the manufacturer's protocol except: 1. Ligation time was increased to 180 minutes, 2. Custom adaptors were used instead of Illumina TruSeq adaptors. The adaptors that were ligated had the following sequences: Adaptor 1: ACACTCTTTCCCTACACGACGCTCTTCCGATCCCC*T and Adaptor 2: /5'Phos/GGGGATCGGGAGAGCACACGTCTGAACTCTAGTCAC. After adaptor ligation, the gDNA fragments were amplified using 5 rounds of PCR with KAPA HiFi 2X master mix (KAPA Biosystems). See S4 Table for primer sequences. After PCR, fragmented gDNA was purified and concentrated using TailorMag Purification beads.

pTriggerAdaptor was linearized using SmaI (NEB) and dephosphorylated using rSAP (NEB). Linearization was verified using gel electrophoresis. 25 ng of fragmented gDNA with ligated adaptors was combined with 25 ng of linearized pTriggerAdaptor. The Gibson Assembly Ultra kit (Synthetic Genomics Inc.) was used according to the manufacturer's instructions. High Efficiency 5-alpha Competent *E. coli* (NEB) were transformed with 3 μl of the Gibson mix and plated on four LB plates containing ampicillin (Sigma). We determined that the Gibson cloning efficiency with pTriggerAdaptor was ~1/250, which was used to calculate the amount of insert needed to generate a library at a given target level of coverage. With an average insert size of ~550 bp, a minimum of ~36,354 plasmids would be needed to fully cover the genome at 1X coverage. For the pilot library, one *E. coli* transformation reaction was performed and 4 plates of bacteria were harvested into 200 ml of LB liquid broth containing ampicillin, and incubated at 37˚C for 12–18 hours, shaking. For the final library, Gibson cloning, *E. coli* transformation, and plasmid preparations were performed over four separate days to create four batches of final library plasmids. The final plasmid library represents the combined total of the four batches. For the final library, 25 *E. coli* transformation reactions were performed and 100 plates of bacteria were harvested into LB liquid broth containing ampicillin, at a concentration of ~1,000 colonies per 200 ml. Plasmids were harvested using the GenElute HP plasmid maxiprep kit (Sigma), using 200 ml of culture per plasmid purification column, and ethanol precipitated according to the manufacturer's instructions.

## Illumina sequencing of gDNA fragments and plasmid samples

Fragmented gDNA (with ligated custom adaptors; see 'plasmid library construction'), or plasmid inserts were amplified with KAPA PCR (Kapa Biosystems) using full length TruSeq Illumina primers, as follows: Primer 1: GATCGGAAGAGCACACGTCTGAACTCCAGTCAC [i7]ATCTCGTATGCCGTCTTCTGCTTG and Primer 2: AATGATACGGCGACCACCGAG ATCTACAC[i5]ACACTCTTTCCCTACACGACGCTCTTCCGATCT. Three technical replicates were sequenced for each sample. Samples were sequenced on an Illumina MiSeq at either the DNA Technologies and Expression Analysis Core (University of California, Davis), or on an Illumina MiSeq at SeqMatic, LLC. Illumina sequencing data analysis was performed at the Bioinformatics Core (University of California, Davis).

## Illumina sequencing analysis

Illumina sequencing data was demultiplexed using barcode sequences added during library preparation with Illumina demultiplexing software. All reads were sequenced as PE 151bp runs. The resulting paired reads were then cleaned using HTStream (https://github.com/ibest/HTStream) to 1) remove reads containing PhiX Control Library sequences, 2) trim Illumina sequencing adaptors, and 3) discard primer dimers and other reads where adaptor trimming was unsuccessful.

Cleaned reads were mapped against the *E. histolytica* release 46 genome assembly (downloaded from https://amoebadb.org/common/downloads/release-46/EhistolyticaHM1IMSS/fasta/data/AmoebaDB-46_EhistolyticaHM1IMSS_Genome.fasta) using bwa mem v2.0pre1 [50]. Output from bwa was passed to samtools v1.10 [51] for sorting and indexing, and also to extract reads from for visualization of specific loci. Results from mapping and subsequent analysis were visualized and reviewed in Geneious Prime 2020.0.3 (https://www.geneious.com). Identifying unique fragments and counting their frequency was done using a custom Python script. Briefly, the script iterates over read-sorted BAM records. Read pairs that were mapped to the same contig with proper orientation, a minimum of 146 matched bases, and no more than four bases soft clipped from the 5' end of either read pair were used to identify fragments. Read mapping orientation was used to infer fragment orientation. A counter for each unique fragment was incremented for additional reads supporting the fragment. Reads that were mapped in proper orientation but that did not meet other criteria were compared against the fragment intervals with an incremented count assigned to the highest scoring overlap. As a last step, the script generates summary statistics, an interval table with counts for all samples, and SAF and gff annotation files for each sample.

Output from the Python script was then loaded into R v3.6.1 [52] for further analysis and to generate figures. Briefly, data.table [53] was used to load the full set of intervals, order them by genomic position, and to compute summaries by sample group. Tidyr [54] was used for reformatting data, and most plots were generated with ggplot2 [55]. The vegan package [56] was used for calculating saturation curve values. The GenomicRanges package [57] was used in combination with gene annotation (downloaded from https://amoebadb.org/common/downloads/release-46/EhistolyticaHM1IMSS/gff/data/AmoebaDB-46_EhistolyticaHM1IMSS.gff) to determine and count fragments targeting genes, all reported counts used a minimum overlap of 27bp. Genome coverage plots were generated with karyoploteR [58] and heatmaps were plotted using pheatmap [59]. Scripts and more details about the analysis are available at the GitHub repository associated with this publication: https://github.com/samhunter/Bettadapur-et-al-Entamoeba-histolytica-RNAi-library.

All sequencing data from this project have been deposited in the Sequence Read Archive (SRA) database, under project number PRJNA672229: https://www.ncbi.nlm.nih.gov/bioproject/672229.

**Transfection and clonal lines.**   Library plasmids were transfected into amoebae using Attractene transfection reagent (Qiagen) [60,61]. 50 ml M199 (Gibco medium M199 with Earle's salts, L-glutamine, and 2.2 g/liter sodium bicarbonate, without phenol red), was supplemented with 0.045 g L-Cysteine, 0.3 g HEPES, 0.005 g ascorbic acid (all from Sigma) and the pH was adjusted to 6.8. To 42.5 ml of this medium, 7.5 ml of heat inactivated Adult Bovine Serum (Gemini bio-products) was added to make transfection medium. Amoebae were resuspended in transfection medium at a concentration of $2.5 \times 10^5$ cells/ml, and then 1.8 ml of amoebae, 10 μg plasmid DNA (in 200 μl of transfection medium), and 37.5 μl Attractene reagent (Qiagen) were combined into a 2.0 ml screw-cap cryotube. Cryotubes were incubated horizontally at 35°C for 3 hours. Two cryotubes per transfection were then added to a T25 flask containing 66 ml of fresh TYI media. 24 hours later, 6 μg/ml geneticin (Invitrogen) was added. Stable transfectants were selected and maintained with geneticin at 6 μg/ml.

The efficiency of stable transfection was determined by performing limiting dilution. Three independent experiments were performed. Amoebae were transfected using the process described above, except that 24 hours after transfection, amoebae were harvested and resuspended in TYI at a concentration of $2 \times 10^6$ cells/ml. Amoebae were then serially diluted into 96-well plates, in the presence or absence of 6 μg/ml geneticin. Plates were incubated in Gas-Pak EZ pouch (BD Biosciences) and growth was monitored daily. The plates without Geneticin were used to calculate the plating efficiency. In the plates without geneticin, growth was seen in wells corresponding to approximately 2.5 cells on average. In the plates containing geneticin, growth was seen in wells containing 781 to 3125 amoebae. Taken together, the corrected stable transfection efficiency was ~1/1000 amoebae.

For generation of clonal mutant lines for phenotypic characterization, limiting dilution was performed in 96-well plates in a GasPak EZ pouch (BD Biosciences). For identification of mutants with growth defects, clonal lines that visually appeared to have slow growth when examined using microscopy were selected for further analysis. RNA was extracted using the Direct-zol RNA kit (Zymo). gDNA (which contains exogenous plasmids) was extracted using the Quick-DNA miniprep kit (Zymo). Sanger sequencing was used to identify the library insert associated with each clonal line. Sequencing was carried out by Genewiz. To identify the gene targeted for knockdown in each clonal line, the insert sequence was used to search the reference genome. Genes with at least 50nt of continuous sequence identity with the insert were considered hits. See S4 Table for sequencing primer information.

## Growth analysis

For growth analysis of RNAi knockdown mutants, clonal stable transfectants (SG1 –SG12) or heterogeneous stable transfectants (SG1i and SG2i) were resuspended to $8 \times 10^3$ cells/ml and 1 ml of cells was added to three tubes containing 13 ml fresh TYI with geneticin. Cells were incubated at 35°C. At days 2, 4, and 6, one tube of cells was harvested per day and counted using a TC20 Automatic cell counter (BioRad). Vector control transfected amoebae were used as a negative control, and superoxide dismutase heterogeneous knockdown mutants were used as a positive control for slow growth. Two technical replicates were performed per experiment and three independent growth curve experiments were performed for each mutant.

For growth analysis of cells overexpressing full-length or truncated SG proteins, geneticin concentration was doubled daily until it reached 48 μg/ml, in order to increase plasmid copy number and lead to protein overexpression. Geneticin concentration was also increased in the

same manner in vector control transfectants. Heterogeneous stable transfectants were resuspended to $5 \times 10^4$ cells/ml and 1 ml of cells was added to five tubes containing 13 ml fresh TYI with 48 µg/ml geneticin. Cells were incubated at 35˚C. At days 1, 2, 3, 4 and 5, one tube of cells was harvested per day and counted using a TC20 Automatic cell counter (BioRad). Vector control transfected amoebae were used as a negative control. Four technical replicates were performed per experiment and three independent growth curve experiments were performed for each cell line. Growth curves were also performed at 24 µg/ml geneticin, which led to the same growth phenotypes seen at higher concentrations of geneticin.

### RT-PCR

RT-PCR was used to assess knockdown in transfectants. RNA was extracted using the Direct-zol RNA kit (Zymo). DNAseI (Turbo) treatment and SuperScript Reverse Transcription II (Invitrogen) was performed as per manufacturer instructions to obtain cDNA. RT-PCR was performed on cDNA harvested from transfectants containing the following constructs: BAR/SH3 domain-containing protein with and without Illumina adaptors, Superoxide Dismutase; SG1/SG1 independent knock down; and SG2/SG2 independent knockdown. GAPDH (EHI_187020) RT-PCR was used as a positive control. See S4 Table for RT-PCR primer sequences.

### Immunofluorescence assays

For immunofluorescence studies, geneticin concentration was doubled daily until it reached 24 µg/ml, in order to increase plasmid copy number and lead to protein overexpression. Amoebae were washed in M199s and fixed with 4% paraformaldehyde in PBS for 30 minutes at room temperature. Samples were permeabilized with 0.2% Triton X-100, or were not permeabilized. For permeabilized samples, for SG1, all subsequent blocking, antibody incubation and wash steps included 0.1% Triton X-100. For permeabilized samples, for SG2, all subsequent blocking, antibody incubation and wash steps included 0.1% Tween 20. For non-permeabilized samples, all subsequent blocking, antibody incubation and wash steps included 0.1% Tween 20. Samples were blocked with goat serum (Jackson ImmunoResearch Laboratories, Inc.) and bovine serum albumin (Gemini). Antibody labeling was carried out in solution. For SG1, a rabbit anti-FLAG primary antibody (Abcam Rabbit Anti-DDDDK tag, binds to FLAG tag sequence, antibody ab1162) was used, followed by an alpaca anti-rabbit Alexa 647 secondary antibody (Jackson ImmunoResearch Laboratories, Inc.). For SG2, a mouse anti-FLAG primary (Sigma Mouse Anti-FLAG M2 monoclonal antibody) was used, followed by an alpaca anti-mouse Cy5 antibody (Jackson ImmunoResearch Laboratories). Samples were stained with DAPI at 40 µg/ml. Data were collected using an ImageStreamX Mark II flow cytometer, equipped with a 40X objective. 10,000 events were collected for each sample and data were analyzed using Amnis IDEAS software (S7 Fig).

For detection of the endogenous Gal/GalNAc lectin, amoebae transfected with vector control (pEhEx) were washed and resuspended in M199s medium and labeled with CellTracker green 5-chloromethylfluorescein diacetate (CMFDA; Invitrogen) at 186 ng/ml for 10 minutes at 35˚C. Samples were fixed and processed as above, without permeabilization. The *E. histolytica* Gal/GalNAc lectin antibody (Clone 7F4; a gift from William A. Petri, Jr., University of Virginia) was used, followed by a Cy5 AffiniPure Goat Anti-Mouse secondary antibody (Jackson ImmunoResearch Laboratories, Inc.). Amoebae were imaged and analyzed as above.

### Statistical analysis

GraphPad Prism was used for statistical analyses. Raw growth analysis data were analyzed using a two-way ANOVA with a post-hoc Dunnett's multiple comparisons test; multiplicity-

adjusted *p*-values are shown. Normalized growth analysis data were analyzed using student's unpaired *t* test statistical analysis and normalization (baseline removal). Mean values and standard deviations are shown in the figures, with *p*-values reported as follows: ns = $p > 0.05$, * = $p < 0.05$, ** = $p < 0.01$, *** = $p < 0.001$, **** = $p < 0.0001$.

## Supporting information

**S1 Fig. Preparation of RNA-free gDNA from *E. histolytica*.** Gel electrophoresis analysis of gDNA extracted using the Wizard gDNA extraction kit (Promega), Quick-DNA miniprep kit (Zymo), or CTAB extraction. **a,** gDNA isolated using the Wizard kit had no apparent RNA contamination, and the gDNA appeared to be of high quality. **b,** gDNA isolated using the Quick-DNA kit or CTAB was contaminated with RNA. RNA contamination was removed by treatment with RNAseIF and subsequent purification. **c,** Bioanalyzer traces of pilot and final gDNA samples, after fragmentation and size selection.
(TIFF)

**S2 Fig. Fragment Mapping Analysis.** Example view of mapped genomic fragments ("intervals") represented in the final plasmid library. For comparison, the mapped paired end reads that give rise to each fragment are shown below each fragment. Shown are results from final plasmid batch two, sequencing replicate one, mapped to contig DS571153.
(TIFF)

**S3 Fig. Appropriate sequencing depth and lack of bias during library construction. a,** Dinucleotide frequency analysis of the HM1:IMSS reference genome, the pilot and final gDNA fragments, and the pilot and final plasmid libraries. Stacked bar plots show the percentage of each dinucleotide. The dinucleotide composition of the gDNA fragments and the final plasmid library is the same as the genome. **b–c,** Rarefaction curves for pilot (panel b) and final (panel c) samples. gDNA fragment samples are shown in grey and plasmid samples are shown in orange. The number of fragments detected with increasing sequence reads shows if sequencing was performed to saturation. Three sequencing replicates are shown for pilot gDNA samples and three sequencing replicates are shown for pilot plasmid samples. Three sequencing replicates are shown for final gDNA samples and three sequencing replicates are shown for each of four batches of final plasmid samples.
(TIFF)

**S4 Fig. Reproducible fragment and read counts from biological and technical replicates.** "Batches" of the final plasmid library were cloned on different days, and represent biological replicates. "Reps" correspond to independently prepared sequencing preparations, and represent technical replicates. **a,** Box plot of read counts per fragment in plasmid samples, with pilot plasmids in orange and final plasmids in grey. $Log_2$ read counts are shown; outliers (outside of 2nd and 98th percentile) are plotted as points, and the box represents the 25th, and 75th percentile with median indicated as a bold line. Most fragments were represented by multiple sequenced reads. **b,** Stacked bar plots of the total number of fragments detected per sample shows that replicates are similar to each other in the number of fragments identified and that roughly equal numbers of forward and reverse fragments were produced consistently. Forward fragments are in the same orientation as genes that are in the forward orientation in the reference genome. Reverse fragments are in the same orientation as genes that are in the reverse orientation in the reference genome.
(TIFF)

**S5 Fig. Annotated genes that were missed or were least covered by sequencing reads.** Heat map showing the number of unique fragments per gene, for the 50 least covered genes in the pilot and final gDNA fragments, and the pilot and final plasmid libraries. The color intensity indicates $Log_2$ fragment coverage, with dark blue indicating no coverage. Stranded fragments are in the same orientation as genes. Unstranded fragments are the total number of fragments, both in the same orientation as genes and in the opposite orientation as genes.
(TIFF)

**S6 Fig. Clonal transfectants contain only one library plasmid.** SG1 gDNA was extracted using the Quick-DNA miniprep kit and PCR amplified using the CS5' and CS3' primers that sit in the pTriggerAdaptor backbone. Sanger sequencing was performed using the RNAi Sequencing Library F primer. Adaptor 2 starts at the "GGGG" series of nucleotides, and upstream of this is the SG1 (EHI_194310) sequence. Noncompeting peaks in the chromatogram suggest that there is one unique genome fragment present in this mutant.
(TIFF)

**S7 Fig. Gating scheme for imaging flow cytometry analysis.** Shown is the gating strategy that was used to analyze imaging flow cytometry data, with the percentage of gated events, and number of gated events, shown below each plot. This example illustrates the gating of the wild-type sample shown in Fig 7. 10,000 events were collected. **a,** In-focus events were gated using a gradient of brightfield (BF). **b,** Events gated in panel a were refined to remove images containing more than one amoeba, by examining the area and the aspect ratio of the object(s) masked in the brightfield image. **c,** Events gated in panel b were refined to remove images that contained saturation in the FLAG channel (channel 11), by examining the raw max pixel intensity (i.e., the intensity of the brightest pixel in the image). **d,** Events gated in panel c as "unsaturated" were further examined to identify images that contained both FLAG and DAPI signal, by examining the intensity of the object masked in the DAPI and FLAG channels. **e,** Events gated in panel d as "+DAPI/FLAG" were further examined to characterize the degree of colocalization (bright detail similarity) between the DAPI and FLAG signals.
(TIFF)

**S8 Fig. Image analysis for FLAG-SG1 transfectants demonstrates that FLAG staining is above background in both permeabilized and non-permeabilized cells.** Amoebae were stably transfected with a plasmid for expression of full-length FLAG-tagged SG1. Immunofluorescence was used to determine the localization of SG1 in heterogeneous transfectants and imaging flow cytometry was used for analysis. The mean fluorescence intensity (MFI) is indicated on each histogram. **a,** Intensity of FLAG staining in FLAG-SG1 transfectants (left panel) and control staining of wild-type cells (right panel). Samples were permeabilized prior to antibody staining. **b,** Intensity of FLAG staining in FLAG-SG1 transfectants (left panel) and control staining of wild-type cells (right panel). Samples were not permeabilized prior to antibody staining.
(TIFF)

**S9 Fig. Gal/GalNAc lectin immunofluorescence and image analysis.** Amoebae were stably transfected with a vector control plasmid (pEhEx) and stained with the cytoplasmic dye CMFDA. Immunofluorescence was used to determine the localization of the endogenous Gal/GalNAc lectin, and imaging flow cytometry was used for analysis. **a,** Localization of the Gal/GalNAc lectin (left panel) and control staining of samples in which primary antibody was omitted (right panel). Samples were not permeabilized prior to antibody staining. Six random cells are shown for each condition. Shown from left to right are bright field images (BF), CMFDA staining (CMFDA, green), and mouse anti-Gal/GalNAc antibody staining (Lectin,

red). The numbers in the BF images indicate the object/image number. **b,** Histograms showing the intensity of the Gal/GalNAc lectin staining (left panel) or control staining of samples in which primary antibody was omitted (right panel). The mean fluorescence intensity (MFI) is indicated on each histogram.
(TIFF)

**S10 Fig. Image analysis for FLAG-SG1 transfectants demonstrates that FLAG staining is above background in permeabilized cells only.** Amoebae were stably transfected with a plasmid for expression of full-length FLAG-tagged SG2. Immunofluorescence was used to determine the localization of SG2 in heterogeneous transfectants and imaging flow cytometry was used for analysis. The mean fluorescence intensity (MFI) is indicated on each histogram. **a,** Intensity of FLAG staining in FLAG-SG2 transfectants (left panel) and control staining of wild-type cells (right panel). Samples were permeabilized prior to antibody staining. **b,** Intensity of FLAG staining in FLAG-SG2 transfectants (left panel) and control staining of wild-type cells (right panel). Samples were not permeabilized prior to antibody staining.
(TIFF)

**S11 Fig. Nuclear cross-reactivity of the mouse anti-FLAG antibody is only seen in permeabilized cells.** Nuclear cross-reactivity of the mouse anti-FLAG antibody was not seen in non-permeabilized cells, supporting that the sample preparation steps did not inadvertently permeabilize samples, and further supporting the surface staining patterns seen in non-permeabilized FLAG-SG1 cells. Colocalization analysis of FLAG and DAPI staining is shown. The percentage of DAPI+/FLAG+ images with colocalization of the two markers are indicated. Amoebae were stably transfected with a plasmid for expression of full-length FLAG-tagged SG2, or wild-type cells were analyzed as a control. a, Colocalization analysis of FLAG and DAPI staining in FLAG-SG2 transfectants (left panel) or wild-type cells (right panel). Samples were permeabilized prior to antibody staining. b, Colocalization analysis of FLAG and DAPI staining in FLAG-SG2 transfectants (left panel) or wild-type cells (right panel). Samples were not permeabilized prior to antibody staining.
(TIFF)

**S12 Fig. Identification of slow growing RNAi knockdown mutants.** Growth analysis of mutants SG1 –SG12. **a, c, e,** Growth analysis demonstrates that selected clonal knockdown mutant lines have significant growth defects compared to vector control transfectants. **b, d, f**, Growth was normalized and compared to vector control transfectants on each day. Mutants exhibited statistically significant growth defects relative to vector control transfectants.
(TIFF)

**S1 Table. Sequencing and fragment count statistics.** Total number of paired reads mapping to the same contig (mappedPairs) and the number of fragments inferred from these mapped pairs for each sample. Reads were mapped with bwa mem, and processed from the resulting SAM formatted file using a custom Python script. Three sequencing replicates were performed for pilot gDNA samples and three sequencing replicates were performed for pilot plasmid samples. Three sequencing replicates were performed for final gDNA samples and three sequencing replicates were performed for each of four batches of final plasmid samples.
(TIF)

**S2 Table. Expression data for annotated genes that were missing in the final plasmid library.** Available empirical expression data for genes that were missing in the final plasmid library (genes that are represented in blue in S5 Fig). Accession numbers, annotation information, and the number of paralogs in the reference genome are indicated. Presence (+) or

absence (-) in gDNA and plasmid datasets is shown. Gene expression from available RNAseq datasets [45,46] is indicated, as the FPKM + 1 value. The highest value for FPKM + 1 for each gene is shown. The total number of spectra that have been identified in proteomics studies is indicated.
(TIF)

**S3 Table. Expression data for representative genes present in the final plasmid library.** To provide a frame of reference for the expression data in S2 Table, shown are the available empirical expression data for 10 random, representative genes that were present in the final plasmid library. Accession numbers, annotation information, and the number of paralogs in the reference genome are indicated. Presence (+) or absence (-) in gDNA and plasmid datasets is shown. Gene expression from available RNAseq datasets [45,46] is indicated, as the FPKM + 1 value. The highest value for FPKM + 1 for each gene is shown. The total number of spectra that have been identified in proteomics studies is indicated.
(TIF)

**S4 Table. Primer sequences.** Sequences of the primers used to generate and sequence plasmids, perform RT-PCR, and create the plasmid RNAi library.
(TIF)

## Acknowledgments

We thank Scott Dawson and Su-Ju Lin for helpful discussions. We thank Lutz Froenicke for helpful advice on designing the library to enable Illumina sequencing. DNA sequencing was carried out by the DNA Technologies and Expression Analysis Cores at the UC Davis Genome Center.

## Author Contributions

**Conceptualization:** Katherine S. Ralston.

**Data curation:** Samuel S. Hunter.

**Formal analysis:** Akhila Bettadapur, Samuel S. Hunter, Rene L. Suleiman, Maura C. Ruyechan, Wesley Huang, Charles G. Barbieri, Hannah W. Miller, Katherine S. Ralston.

**Funding acquisition:** Katherine S. Ralston.

**Investigation:** Akhila Bettadapur, Rene L. Suleiman, Maura C. Ruyechan, Wesley Huang, Charles G. Barbieri, Hannah W. Miller, Tammie S. Y. Tam, Katherine S. Ralston.

**Project administration:** Katherine S. Ralston.

**Resources:** Matthew L. Settles.

**Software:** Samuel S. Hunter.

**Supervision:** Matthew L. Settles, Katherine S. Ralston.

**Visualization:** Akhila Bettadapur, Samuel S. Hunter, Rene L. Suleiman, Hannah W. Miller, Katherine S. Ralston.

**Writing – original draft:** Akhila Bettadapur, Samuel S. Hunter, Katherine S. Ralston.

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
