## [Decision Letter · Decision Letter 0]

26 Sep 2021

Dear Katy,

Thank you very much for submitting your manuscript "Establishment of quantitative RNAi-based forward genetics in Entamoeba histolytica and identification of genes required for growth" for consideration at PLOS Pathogens. As with all papers reviewed by the journal, your manuscript was reviewed by members of the editorial board and by several independent reviewers. The reviewers appreciated the attention to an important topic. Based on the reviews, we are likely to accept this manuscript for publication, providing that you modify the manuscript according to the review recommendations.

You will note that one reviewer requested extensive major revisions including expansion of the study. We understand that this is beyond the scope of this manuscript and do not expect those experiments to be done. Despite these concerns we agree with Reviewers 1 and 2 that the impact of this work is substantial and important to move the field forward. In a revised manuscript, please address the minor comments by all reviewers, either experimentally or in the response/revised manuscript.

Sincerely,

Upinder Singh

Guest Editor

PLOS Pathogens

Vern Carruthers

Section Editor

PLOS Pathogens

Kasturi Haldar

Editor-in-Chief

PLOS Pathogens

orcid.org/0000-0001-5065-158X

Michael Malim

Editor-in-Chief

PLOS Pathogens

orcid.org/0000-0002-7699-2064

Reviewer Comments (if any, and for reference):

Reviewer's Responses to Questions

**Part I - Summary**

Reviewer #1: In this study, Bettadapur and colleagues describe the first forward-genetics RNAi screen in the human protozoan pathogen, Entamoeba histolytica. This pathogen is a globally-important causative agent of dysentery and liver abscess. There is no vaccine and currently available drugs are fraught with side effects. Thus, any molecular and or cellular insight into this pathogen may reveal new targets for therapy. This study provides a new tool for the community of researchers to identify such targets. The strengths of the study are its novelty and robust analyses of the RNAi libraries. The authors have nicely responded to the previous critiques.

Reviewer #2: Bettadapur et al generate a library of gDNA fragments flanked by an RNAi trigger in order to establish a forward genetics screening system for E. Histolytica. They use this system to screen 80 transgenic cell lines and identified 12 Slow Growth (SG) mutants. They sequenced the inserts to identify the targeted genes and learned that the orientation of the gDNA fragment has no impact on the effectiveness of knockdown. Toward characterizing SG1 and SG2 mutants they localized these proteins.

Reviewer #3: This paper from Bettadapur, et al, takes a very creative approach to establish the first forward genetic gene-silencing system for Entamoeba histolytica that is suitable for conducting genome-wide screens. The authors do a great job describing the approach, building and characterizing a plasmid library, and successfully completing a pilot screen that validates the method. The work appears to have been done to a high standard, and the figures and writing are excellent. As noted on the prior reviews, there is limited biological follow up on "hits" from the pilot screen, although this appears to have been augmented by confirming phenotypes and looking at localization for two of the silenced genes. I did not review the original version of this manuscript and I have to say that I think the importance of the paper is in description and validation of the method. The reality is that this type of method has been sorely needed and has the potential to jump start Entamoeba histolytica research--the lack of magnificent new biological insights from the pilot screen is no surprise, and, given the time and resources that would be needed to conduct a whole-genome screen with thorough follow up for a more interesting phenotype, I personally think it would do the Entamoeba research community (and perhaps people affected by amebiasis) a disservice to insist on more work with a resultant delay in publication. I have minor comments that, if addressed, would strengthen the manuscript.

**Part II – Major Issues: Key Experiments Required for Acceptance**

Reviewer #1: No major issues.

Reviewer #2: 1. The authors constructed a high-quality library, but the utility of this library has not been established. On Line 505 the authors state that complete genome coverage requires 36,354 plasmids. If each isolated transgenic parasite caries a unique plasmid, then the 80 isolated parasites screened in this manuscript represents about 0.22% genome coverage. Screening 80 colonies does not meet the bar for a forward genetics screening system. Consider that knockdown of 80 genes is still well within the range of what is possible for a directed screen, the authors need to devise a means to screen at least several thousand inserts before they can conclude they have a game changing forward genetics system.

2. Why report generating a complete library if the performed screen was from the pilot library? Given that only 80 strains were screened the pilot library already contains far more unique plasmids. The authors need to work on a method to increase the throughput of screening.

3. Slow growth is not a particularly informative phenotype. I imagine the authors have some plan to use this screening strategy to probe a biological process of interest. A demonstration that the screening system could be used to identify mutants that are defective in a cellular process of interest would be more meaningful and impactful.

Reviewer #3: None

**Part III – Minor Issues: Editorial and Data Presentation Modifications**

Reviewer #1: 1. The conclusion that the protein encoded by EHI_194310 is on the cell surface is not well-supported by the immunofluorescence data. There is a “haze” of staining on the “unpermeabilized” cells, but it is difficult to see. Even fixation with paraformaldehyde will cause some permeabilization. Thus, it possible that the haze is simply some of the anti-FLAG antibody getting into slightly permeabilized cells. This would be particularly true if the expression of the FLAG-tagged version of the protein was high. The authors state that they did not observe such surface staining in the EHI_007010 overexpressing cell line, but they did not show the data. This reviewer assumes that such a conclusion was drawn because no haze was observed whatsoever. Perhaps providing the EHI-007010 staining data and showing that it is different from the EHI_194310 staining data might be more convincing. Comparing the image analyses for both staining experiments might also help illustrate the contrast.

2. In the Introduction, the authors provide a nice review and timeline of the development of forward genetics and RNAi techniques for use in this parasite. Recently, a study showing proof of principle of a CRISPR/Cas9 system for use in Entamoeba (Kangussu-Marcolino et al., 2021). It would be fair to include this information/cite this study in the Introduction and/or Discussion.

3. The data in Figure 5 are not uniformly presented. In 5B, two of the panels present the RT data first (+ -), while the third panel presents the RT data second (- +). In 5D, RT data are presented second (- +). This is confusing to the reader.

Reviewer #2: 1. The characterization of SG1 and SG2 is not informative. I am not suggesting the authors do more because this is not my major issue with the work.

2. The choice of Imaging flow cytometry is not justified. The authors could acquire much higher resolution images using a confocal with a high NA 60 or 63X objective with the advantage of being able to see how the protein localization changes at different focal planes within cells. It is not clear that SG1 is in the plasma membrane. It could be in the ER and be pushed up against the cell cortex. Consider using a plasma membrane stain or protein marker already known to be at the plasma membrane.

3. The authors chose to use 500 bp fragments in the library. Clearly this size fragment can work. Could the authors compare a 500 bp vs 1000 bp fragment for their BAR/SH3 control gene to give us some sense for how their choice to use 500bp fragments in their library might impact efficacy of silencing?

4. How stable is the silencing phenotype? Does the level of silencing change over time? This would be an important consideration for how long strains can be maintained before being used for screening.

5. Will gDNA fragments that hit promoters silence the associated gene?

6. Figures 6 and 7 lack scale bars.

Reviewer #3: Minor concerns :

1) Author summary line 53: "enable forward genetics for the first time..."--as referenced by the authors, this is not quite true, since at least two prior "forward genetic" studies have been performed using overexpression libraries. Maybe better as "...the RNAi pathway to couple gene silencing with forward genetics for the first time..."

2) Introduction line 73: Despite the cited Cochrane Review, the statement "therapy is limited to nitroimidazoles" is incorrect--nitazoxanide is an effective alternative treatment for amebiasis. Maybe change to simply "therapy is limited".

3) Results line 134 "a library of genes..." would be more accurate as "a library of gene fragments" or "genomic fragments".

4) Line 138: use of the word "mutants" here and in figure 1b is not accurate--these are not actually mutants. Although a little awkward, I think better as "targeted genes" or "silenced genes" throughout.

5) Line 263 and throughout: I don't agree that growth rate is what's being measured, as the number of cells reflects both the rate of proliferation (i.e. growth rate) and persistence/longevity of cells in the culture. As the authors note, the increase in DAPI staining with the SG1 mutant indicates more average DNA content, which may indicate a higher rate of proliferation. That said, this conclusion could only be made with a more sophisticated experiment, such as using a vital dye (e.g. CFSE) and measuring the numbers/proportion of parasites that have divided 1, 2, 3x, etc. It would improve the accuracy to simply replace growth rate throughout with "cell number" or similar. If growth rate, then I think best to do a more sophisticated analysis. As above, I believe the importance of the paper is simply the method and library.

6) The methods indicate that t-tests were used to determine statistical significance. I believe that where multiple groups are being compared to the control the appropriate test would be an ANOVA or another test suitable for multiple comparisons.

PLOS authors have the option to publish the peer review history of their article (what does this mean?). If published, this will include your full peer review and any attached files.

Reviewer #1: No

Reviewer #2: No

Reviewer #3: No

Figure Files:

Data Requirements:

Reproducibility:

References:

---

## [Editor Report · Decision Letter 1]

3 Nov 2021

Dear Dr Ralston,

We are pleased to inform you that your manuscript 'Establishment of quantitative RNAi-based forward genetics in Entamoeba histolytica and identification of genes required for growth' has been provisionally accepted for publication in PLOS Pathogens.

Best regards,

Upinder Singh

Guest Editor

PLOS Pathogens

Vern Carruthers

Section Editor

PLOS Pathogens

Kasturi Haldar

Editor-in-Chief

PLOS Pathogens

orcid.org/0000-0001-5065-158X

Michael Malim

Editor-in-Chief

PLOS Pathogens

orcid.org/0000-0002-7699-2064

Many thanks for your efforts in submitting the revised manuscript, which we believe adequately addresses the reviewer comments for the majority of reviewers.
---

## [Editor Report · Acceptance letter]

23 Nov 2021

Dear Dr Ralston,

We are delighted to inform you that your manuscript, "Establishment of quantitative RNAi-based forward genetics in </i>Entamoeba histolytica </i> and identification of genes required for growth," has been formally accepted for publication in PLOS Pathogens.

Best regards,

Kasturi Haldar

Editor-in-Chief

PLOS Pathogens

orcid.org/0000-0001-5065-158X

Michael Malim

Editor-in-Chief

PLOS Pathogens

orcid.org/0000-0002-7699-2064